# *Rank-Learner*: Orthogonal Ranking of Treatment Effects

Henri Arno [1]   Dennis Frauen [2 3]   Emil Javurek [2 3]   Thomas Demeester [1]   Stefan Feuerriegel [2 3]

## Abstract

Many decision-making problems require ranking individuals by their treatment effects rather than estimating the exact effect magnitudes. Examples include prioritizing patients for preventive care interventions, or ranking customers by the expected incremental impact of an advertisement. Surprisingly, while causal effect estimation has received substantial attention in the literature, the problem of directly learning *rankings of treatment effects* has largely remained unexplored. In this paper, we introduce *Rank-Learner*, a novel two-stage learner that directly learns the ranking of treatment effects from observational data. We first show that naïve approaches based on precise treatment effect estimation solve a harder problem than necessary for ranking, while our *Rank-Learner* optimizes a pairwise learning objective that recovers the true treatment effect ordering, without explicit CATE estimation. We further show that our *Rank-Learner* is Neyman-orthogonal and thus comes with strong theoretical guarantees, including robustness to estimation errors in the nuisance functions. In addition, our *Rank-Learner* is model-agnostic, and can be instantiated with arbitrary machine learning models (e.g., neural networks). We demonstrate the effectiveness of our method through extensive experiments where *Rank-Learner* consistently outperforms standard CATE estimators and non-orthogonal ranking methods. Overall, we provide practitioners with a new, orthogonal two-stage learner for ranking individuals by their treatment effects.

## 1. Introduction

Across many application domains, decision-makers must rank individuals by their treatment effects (Kamran et al., 2024). Such ranking problems arise especially when limited resources make it necessary to prioritize those who benefit most from the treatment.

**Examples.** *In healthcare, clinicians need to triage patients according to who should receive intensive care during periods when demand exceeds capacity (Aquino et al., 2022; Vinay et al., 2021) or prioritize patients for preventive care (Kraus et al., 2024). In marketing, firms must decide which customers to target with retention offers or which prospects to reach with costly advertisements (Devriendt et al., 2021; Gharibshah & Zhu, 2021). In public policy, governments must decide which individuals to target with policy interventions (Card et al., 2018). In all of these settings, effective decision-making depends on the relative ordering of treatment effects rather than on their exact magnitudes.*

Interestingly, the task of learning-to-rank individuals by their treatment effects from observational data has received relatively little attention (see Section 2), especially when compared to the extensive literature on estimating heterogeneous treatment effects. One reason is that standard learning-to-rank methods (Liu, 2011; Cao et al., 2007; Burges et al., 2005) are **not** directly applicable in this setting; such methods would require supervision via the relative ordering of treatment effects, but treatment effects are never directly observed in observational data (Rubin, 2005).

Instead, existing works aimed at ranking individuals by their treatment effects proceed *indirectly* by first estimating conditional average treatment effects (CATEs) from observational data (Künzel et al., 2019; Wager & Athey, 2018). Concretely, a naïve approach is to first estimate CATEs using state-of-the-art methods (e.g., two-stage learners such as the DR- and R-learners (Kennedy, 2023; Nie & Wager, 2021)) and then rank individuals according to their estimated effects. However, as we show later, this strategy solves a harder learning problem than necessary for ranking. In contrast, only a few works have explored how to learn treatment effect rankings *directly* (Kamran et al., 2024; Vanderschueren et al., 2024), but **none** of these methods are Neyman-orthogonal and thus lack favorable theoretical properties (e.g., robustness to nuisance estimation error).

[1]Ghent University - imec, Ghent [2]LMU Munich, Munich [3]Munich Center for Machine Learning (MCML). Correspondence to: Henri Arno <henri.arno@ugent.be>.

*Proceedings of the 43rd International Conference on Machine Learning*, Seoul, South Korea. PMLR 306, 2026. Copyright 2026 by the author(s).

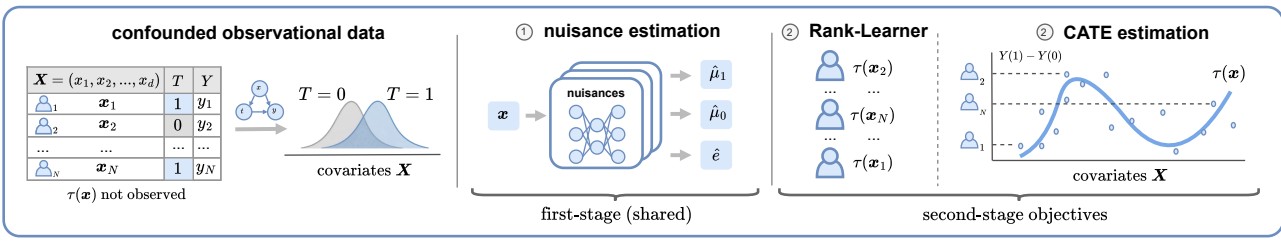

*Figure 1.* **Two-stage learners for ranking treatment effects.** *Left:* Confounded observational data with unobserved treatment effects. *Center:* First-stage estimation of nuisance functions (i.e., response surfaces and propensity score). *Right:* Second-stage objectives: our *Rank-Learner* vs. standard CATE estimation. Here, our proposed *Rank-Learner* directly optimizes a pairwise, Neyman-orthogonal ranking objective that targets the relative ordering of treatment effects. In contrast, standard CATE estimation optimizes a pointwise regression objective to recover treatment effect magnitudes, which is harder than necessary for ranking.

In this paper, we introduce ***Rank-Learner***, a novel two-stage learner for *ranking individuals by their treatment effects from observational data*. Rather than first estimating CATEs and then ranking individuals by their magnitudes, our *Rank-Learner* directly targets the ranking task itself by optimizing a population-level pairwise ranking objective, without explicit CATE estimation. This avoids solving the unnecessarily difficult problem of recovering precise treatment effects and instead focuses directly on the relevant quantity in our task, namely, the relative ordering of treatment effects between individuals (see Figure 1). Our *Rank-Learner* follows a two-stage approach: in the first stage, nuisance functions are estimated using flexible machine learning models; in the second stage, these estimates are plugged into a *novel orthogonal pairwise loss* to learn the ranking function. As a result, *Rank-Learner* is fully model-agnostic and can thus be instantiated with arbitrary machine learning models, such as neural networks. The orthogonality of our loss allows for favorable theoretical properties and ensures robustness to estimation errors in the nuisance functions. To the best of our knowledge, our *Rank-Learner* is the first orthogonal two-stage learner for ranking treatment effects.

Our main **contributions** are:[1] **(1)** We propose *Rank-Learner*, a novel two-stage learner for directly learning rankings of treatment effects from observational data. **(2)** We prove that the learning objective inside our *Rank-Learner* is Neyman-orthogonal. **(3)** We demonstrate that our *Rank-Learner* outperforms standard CATE estimators and non-orthogonal rankers across extensive experiments.

## 2. Related work

We now review related work on (i) learning-to-rank, (ii) CATE estimation, and (iii) recent methods on ranking individuals by their treatment effects. An extended review on policy learning and uplift modeling is given in Appendix A.

[1]Code is available at https://github.com/henriarnoUG/rank-learner.

**Learning-to-rank.** Learning-to-rank refers to supervised learning methods for learning an ordering over a set of items (Liu, 2011). These approaches originate from information retrieval, where the goal is typically to rank documents for a given query based on relevance judgments. Existing methods can be classified into three broad categories depending on the type of supervision used during training: (i) *pointwise methods* define a loss using item-level relevance labels; (ii) *pairwise methods* rely on relevance comparisons between pairs of items and learn relative preferences (e.g., Burges et al., 2005); and (iii) *listwise methods* are supervised using entire ranked lists or permutations of items by relevance (e.g., Buyl et al., 2023; Cao et al., 2007). However, learning-to-rank methods are **not** directly applicable in our setting because the required supervision is unavailable since treatment effects are never directly observed.

**CATE estimation.** There is an extensive literature on estimating heterogeneous treatment effects from observational data, often formalized by the *conditional average treatment effect* (CATE). At a high level, existing approaches can be classified as *model-based* or *model-agnostic*. Model-based methods rely on specific machine learning models designed for treatment effect estimation, such as causal forests (Wager & Athey, 2018) or specialized neural networks (Shalit et al., 2017). Model-agnostic methods, also referred to as two-stage learners or meta-learners, define generic estimation procedures that can be instantiated with arbitrary machine learning models (Künzel et al., 2019).

State-of-the-art CATE estimators are *Neyman-orthogonal* two-stage learners, including the DR- and R-learners (Kennedy, 2023; Nie & Wager, 2021). In the first stage, so-called *nuisance functions* are estimated, which capture components of the data-generating process, like the treatment propensity and the conditional outcomes. In the second stage, treatment effects are estimated by optimizing a learning objective constructed from these nuisance estimates. Neyman-orthogonality ensures robustness of the second-stage model to estimation errors in the nuisance functions (Foster & Syrgkanis, 2023). However, the above methods

are designed to estimate the CATE *magnitudes*, but are **not** designed to directly learn a *ranking* of the CATEs.

**Ranking of treatment effects.** Recently, two works have explicitly considered the problem of ranking individuals by their treatment effects. Kamran et al. (2024) propose a tree-based method that directly optimizes a non-differentiable ranking criterion based on doubly robust pseudo outcomes (we refer to this method as *tree ranker*). In contrast, Vanderschueren et al. (2024) do not propose a new ranking method, but empirically evaluate different combinations of standard CATE estimators and conventional learning-to-rank objectives (our *plug-in ranker* baseline closely follows this strategy). However, **neither** approach is Neyman-orthogonal, implying that the resulting learning objectives are sensitive to estimation errors in the nuisance functions.

**Research gap.** Taken together, existing approaches either focus on estimating CATE magnitudes, or learn treatment-effect rankings using non-orthogonal learning objectives or model-specific classes. As summarized in Table 1, this leaves a gap for model-agnostic two-stage learners that directly target ranking, while remaining Neyman-orthogonal.

| Method | Rank. | Agn. | Orth. | Reference |
|---|---|---|---|---|
| T-learner | ✗ | ✓ | ✗ | Künzel et al. (2019) |
| DR-learner | ✗ | ✓ | ✓ | Kennedy (2023) |
| Tree ranker | ✓ | ✗ | ✗ | Kamran et al. (2024) |
| Plug-in ranker | ✓ | ✓ | ✗ | cf. Vanderschueren et al. (2024) |
| ***Rank-learner*** (ours) | ✓ | ✓ | ✓ | *This work* |

*Table 1.* **Positioning of methods.** *Rank.*: directly targets treatment effect ranking (rather than CATE magnitudes). *Agn.*: model-agnostic method that can be instantiated with arbitrary machine learning models. *Orth.*: Neyman-orthogonal learning objective with respect to nuisance estimation. T- and DR-learner are representative examples of two-stage learners for CATE estimation. Main experiments focus on two-stage learners, a comparison to *tree ranker* is given in Appendix F.4 (Table 12).

## 3. Problem setup

**Data.** We consider a standard causal inference setting with observations $W = (X, T, Y) \sim \mathbb{P}$, where $X \in \mathcal{X} \subseteq \mathbb{R}^d$ represents covariates, $T \in \{0, 1\}$ is a binary treatment, and $Y \in \mathbb{R}$ is a continuous outcome. We assume that we have a dataset $\{x_i, t_i, y_i\}_{i=1}^n$ from $n \in \mathbb{N}$ individuals, sampled i.i.d. from $\mathbb{P}$. We build upon the potential outcomes framework (Rubin, 2005), meaning that each individual in the population has two potential outcomes, $Y(1)$ and $Y(0)$, where $Y(t)$ represents the outcome that we would observe had the treatment been $T = t$. Then, the CATE is defined as

$$\tau(x) = \mathbb{E}[Y(1) - Y(0) \mid X = x]. \quad (1)$$

**Identification.** Since potential outcomes are not directly observed, we rely on standard identification assumptions under which the CATE can be expressed in terms of the observed data (Imbens & Rubin, 2015; Rosenbaum & Rubin, 1983).

**Assumption 3.1** (Standard causal inference assumptions). For all $t \in \{0, 1\}$ and $x \in \mathcal{X}$, it holds: (i) *Consistency*: The observed outcome equals the potential outcome under the received treatment: $Y = Y(t)$ when $T = t$. (ii) *Positivity*: Each individual has a non-zero probability of receiving either treatment: $0 < e(x) < 1$ where $e(x) = P(T = 1 \mid X = x)$ is the propensity score. (iii) *Unconfoundedness*: There are no unmeasured confounders: $Y(t) \perp\!\!\!\perp T \mid X$.

Under Assumption 3.1, the CATE is identified as

$$\tau(x) = \mu_1(x) - \mu_0(x), \quad (2)$$

where $\mu_t(x) = \mathbb{E}[Y \mid T = t, X = x]$ are the conditional outcome regressions, referred to as *response surfaces*. Together with the propensity score $e(x)$, these response surfaces $\mu_1(x)$ and $\mu_0(x)$ constitute the *nuisance functions*, which we denote collectively as $\eta = (\mu_1, \mu_0, e)$.

**Objective.** In this work, we focus on ranking individuals by their treatment effects $\tau(x)$. To this end, we consider a real-valued *scoring function* $g : \mathcal{X} \to \mathbb{R}$, which is used to rank individuals based on their covariates. We require that the ordering of $g(x)$ agrees with that of $\tau(x)$, meaning that, for any $x, x' \in \mathcal{X}$, we have

$$g(x) > g(x') \quad \text{whenever} \quad \tau(x) > \tau(x'). \quad (3)$$

Equivalently, any scoring function of the form $g(x) = h(\tau(x))$, with $h$ strictly increasing, induces the same ordering as $\tau(x)$. Throughout the paper, we assume that ties in $\tau(x)$ do not occur. Note that this task involves only ranking, and the magnitudes of $\tau(x)$ are *not* of direct interest.

## 4. Principles of orthogonal learning

**Bias of plug-in rankers.** A simple approach is to use *plug-in rankers*, which proceed by first estimating the nuisance functions $\hat{\mu}_t(x)$ (the response surfaces) from the observed data, and plugging these estimates into the identification formula from Eq. (2) to obtain $\hat{\tau}(x) = \hat{\mu}_1(x) - \hat{\mu}_0(x)$. These treatment effect estimates can then be used to construct the supervision targets for any learning-to-rank objective (examples discussed below), and $g$ is fit by minimizing the corresponding empirical loss $\mathcal{L}^{\text{plug}}(g, \hat{\eta})$ (where $\hat{\eta}$ collects the estimated response surfaces). However, it is well established that such plug-in approaches suffer from *plug-in bias* (Kennedy, 2024): estimation errors in the nuisance functions enter the training objective directly, and spill over into the fitted scoring function $g$.

**Neyman-orthogonality.** Plug-in bias can be addressed by developing learning objectives that are *Neyman-orthogonal* with respect to the nuisance functions. Concretely, one first

estimates the nuisances $\hat{\eta}$ and then fits the target model by minimizing an objective $\mathcal{L}^{\mathrm{corr}}(g, \hat{\eta})$ designed to be first-order insensitive to nuisance error. Formally, a loss is *orthogonal* if, for any perturbation directions $\Delta g$ and $\Delta \eta$,

$$D_\eta D_g \, \mathcal{L}^{\mathrm{corr}}(g^0, \eta^0)[\Delta g, \, \Delta \eta] = 0 \qquad (4)$$

where $D_\eta$ and $D_g$ denote directional derivatives, $\eta^0$ are the true nuisance functions and $g^0$ is a population minimizer of $\mathcal{L}^{\mathrm{corr}}(g, \eta^0)$ (Foster & Syrgkanis, 2023; Chernozhukov et al., 2018). Intuitively, this property means that the gradient of the loss with respect to $g$ is robust against estimation errors in the nuisances. Orthogonality usually implies additional favorable properties, such as fast convergence rates (Foster & Syrgkanis, 2023; Nie & Wager, 2021).

**Constructing orthogonal losses.** A common strategy to obtain a Neyman-orthogonal objective is to start from the plug-in loss $\mathcal{L}^{\mathrm{plug}}(g, \eta)$ and apply a correction based on its *influence function* (Kennedy, 2024; Foster & Syrgkanis, 2023). Intuitively, the influence function measures the first-order sensitivity of the population objective to infinitesimal perturbations of the data-generating distribution $\mathbb{P}$ (for background and details, we refer to Appendix B). Orthogonal learning has recently been explored in novel applications, such as preference-based LLM evaluation (e.g. Frauen et al., 2026). However, to the best of our knowledge, Neyman-orthogonal learning objectives for learning-to-rank treatment effects are currently missing.

## 5. Orthogonal ranking of treatment effects

**Motivation.** In this section, we first show that standard CATE learning solves a harder problem than required for ranking, and motivate a pairwise ranking objective instead. We then introduce a smooth surrogate loss that enables orthogonalization, and present our main contribution: a novel Neyman-orthogonal ranking loss that targets the treatment effect ordering directly. This learning objective forms the basis of our *Rank-Learner* (Section 6).

**Why CATE learning is harder than needed for ranking.** A naïve strategy to learn the treatment effect ordering is to first estimate CATEs and then rank individuals by these estimates. This corresponds to learning a scoring function $g$ by minimizing the mean squared error loss

$$\mathcal{L}^{\mathrm{cate}}(g, \eta) = \mathbb{E}_X\left[\left(g(X) - \tau(X)\right)^2\right], \qquad (5)$$

which is the canonical population objective for standard CATE estimation (Morzywolek et al., 2024). The loss $\mathcal{L}^{\mathrm{cate}}$ is uniquely minimized at $g(x) = \tau(x)$, and therefore requires recovering the *full* CATE function. While this, of course, gives the correct ranking, it is also unnecessarily difficult because we do not need correct treatment effect magnitudes. Below, we relax the optimization objective accordingly.

**Why a ranking loss is preferred.** Since ranking only depends on the *relative* ordering of treatment effects, we consider the pairwise ranking objective (Burges et al., 2005)[2]

$$\mathcal{L}^{\mathrm{bin}}(g, \eta) = \mathbb{E}_{X, X'}\left[\ell\left(p_g(X, X'), \, b_\tau(X, X')\right)\right], \quad (6)$$

where $\ell(p, t) = -t \log p - (1-t) \log(1-p)$ denotes the binary cross-entropy loss (we will use this shorthand notation throughout), and

$$p_g(X, X') = \sigma\left(g(X) - g(X')\right), \qquad (7)$$

$$b_\tau(X, X') = \mathbf{I}\{\tau(X) > \tau(X')\}, \qquad (8)$$

where $\sigma(\cdot)$ is the logistic sigmoid and $\mathbf{I}\{\cdot\}$ the indicator function. Here, $p_g(X, X')$ can be interpreted as a *pairwise preference probability* that encodes the model's confidence that $X$ should be ranked ahead of $X'$. The label $b_\tau(X, X')$ provides the corresponding supervision by indicating whether $X$ has a larger treatment effect than $X'$. Optimizing this loss yields a single scoring function $g$ that can be used to rank individuals directly at inference.

At the population level, the infimum of $\mathcal{L}^{\mathrm{bin}}$ (which is not attainable by any finite $g$) can be approached arbitrarily closely by *any scoring function that preserves the treatment effect ordering*. In particular, any $g(x) = h(\tau(x))$ with $h$ strictly increasing is population-optimal in this sense. This implies invariance to the form $h$, as long as it preserves the effect ordering. In contrast to $\mathcal{L}^{\mathrm{cate}}$, which identifies treatment effect magnitudes, $\mathcal{L}^{\mathrm{bin}}$ imposes only ordering constraints on $g$. Since our goal is to recover only the ordering of treatment effects, we therefore focus on the *easier* loss $\mathcal{L}^{\mathrm{bin}}$ (see Appendix D for details).

**Smooth surrogate ranking loss.** So far, we have assumed oracle access to $\tau(x)$, but, in observational data, the treatment effect $\tau(x)$ is unobserved. Simply replacing $\tau(x)$ with an estimate $\hat{\tau}(x)$ suffers from plug-in bias, motivating an influence-function-based correction (cf. Section 4). However, to orthogonalize $\mathcal{L}^{\mathrm{bin}}$ via the influence function, the loss must depend smoothly on the treatment effects (and more generally, on the nuisance components $\eta$ of the data-generating process). Since $\mathcal{L}^{\mathrm{bin}}$ uses the indicator targets $b_\tau(X, X')$ that are discontinuous, it is non-differentiable with respect to $\tau$. To overcome this, we replace these binary targets with smooth probabilistic surrogates and consider the soft ranking loss (cf. Vanderschueren et al., 2024)

$$\mathcal{L}^{\mathrm{soft}}(g, \eta) = \mathbb{E}_{X, X'}\left[\ell\left(p_g(X, X'), \, t_\tau(X, X')\right)\right], \quad (9)$$

where

$$t_\tau(X, X') = \sigma\left(\frac{\tau(X) - \tau(X')}{\kappa}\right), \qquad (10)$$

---

[2]Throughout, $\mathbb{E}_{X, X'}$ denotes the expectation over i.i.d. draws $X, X' \sim \mathbb{P}(X)$, and $\mathbb{E}_{W, W'}$ over $W, W' \sim \mathbb{P}$ analogously.

and $\kappa > 0$ is a smoothness parameter. These targets can be interpreted as the probability that $X$ has a larger treatment effect than $X'$. The parameter $\kappa$ controls how sharp this comparison is: smaller values push the targets closer to the binary indicators $b_\tau(X, X') \in \{0, 1\}$, larger values produce softer targets closer to $0.5$. Crucially, this smoothness is what allows us to orthogonalize the ranking objective.

**Our novel Neyman-orthogonal ranking loss.** Since the targets in $\mathcal{L}^{\text{soft}}$ are smooth in $\tau$, we can now orthogonalize this loss based on its influence function. Doing so introduces a correction that is expressed in terms of the full observational units $W = (X, T, Y)$, and leads to the following theorem.

**Theorem 5.1** (Neyman-orthogonality)**.** *We define the loss*

$$\mathcal{L}^{\text{orth}}(g, \eta) = \mathbb{E}_{W,W'}\left[\ell\big(p_g(X, X'), \tilde{t}_\eta(W, W')\big)\right], \quad (11)$$

*with pseudo labels*

$$\tilde{t}_\eta(W, W') = t_\tau(X, X') + \omega_\tau(X, X')\,\Delta_\eta(W, W'), \quad (12)$$

*where*

$$\omega_\tau(X, X') = \frac{1}{\kappa}\, t_\tau(X, X')\big(1 - t_\tau(X, X')\big), \quad (13)$$

$$\Delta_\eta(W, W') = \big(\phi_\eta(W) - \tau(X)\big) - \big(\phi_\eta(W') - \tau(X')\big), \quad (14)$$

*with $\phi_\eta(W)$ the doubly robust score*

$$\phi_\eta(W) = \frac{T}{e(X)}\big(Y - \mu_1(X)\big) - \frac{1-T}{1-e(X)}\big(Y - \mu_0(X)\big)$$
$$+ \mu_1(X) - \mu_0(X). \quad (15)$$

*Then the loss $\mathcal{L}^{\text{orth}}(g, \eta)$ is Neyman-orthogonal with respect to the nuisance components $\eta = (\mu_1, \mu_0, e)$.*

*Proof.* See Appendix C.

Theorem 5.1 shows that the proposed objective $\mathcal{L}^{\text{orth}}(g, \eta)$ is first-order insensitive to nuisance estimation error, but we need to verify that it still targets the correct treatment effect ordering. Therefore, we characterize its population minimizers evaluated at the true nuisance functions.

**Theorem 5.2** (Minimizers of the orthogonal loss)**.** *Let $\eta^0$ denote the true nuisance functions, and let $\tau^0$ be the corresponding CATE. For any fixed and finite $\kappa > 0$, the orthogonal ranking loss $\mathcal{L}^{\text{orth}}(g, \eta^0)$ is minimized by any scoring function of the form*

$$g(x) = \frac{1}{\kappa}\tau^0(x) + c, \quad (16)$$

*for some constant $c \in \mathbb{R}$, and therefore recovers the treatment effect ordering.*

*Proof.* See Appendix D.

Together, Theorem 5.1 (Neyman-orthogonality) and Theorem 5.2 (correct population minimizers)[3] establish that $\mathcal{L}^{\text{orth}}(g, \eta)$ targets the correct treatment effect ordering while remaining robust to nuisance estimation errors. Table 2 summarizes the four considered learning objectives, their population optima, the induced constraints on $g$, and whether they are Neyman-orthogonal.

**Intuition of the pseudo labels.** Intuitively, the pseudo labels in Eq. (12) can be viewed as the soft ranking targets augmented with an orthogonal correction. Note that the correction term itself is the product of *(i)* an uncertainty-dependent weight $\omega_\tau$ and *(ii)* a difference of doubly robust scores $\Delta_\eta$. Hence, when the (plug-in) soft target is close to 0 or 1, the implied pairwise ordering of treatment effects is non-ambiguous and the weight is small, so the pseudo label remains close to the soft target. In contrast, when the soft target is near $0.5$, the pairwise ordering tends to be ambiguous and the weight becomes large, thus activating the correction and shifting the pseudo label according to the doubly robust score difference.

The smoothness parameter $\kappa$ controls *when* and *how strongly* this correction is applied through the weight in Eq. (13). Here, smaller values of $\kappa$ sharpen the soft targets, pushing them closer to 0 or 1 and reducing the number of pairs for which the correction is active. At the same time, for the remaining ambiguous pairs, the correction weight increases due to the $(1/\kappa)$ scaling. As a result, the orthogonal correction concentrates on fewer, harder pairs, but with a larger impact on their pseudo labels.

**Behavior of the orthogonal ranking loss.** The population minimizers of the orthogonal loss $\mathcal{L}^{\text{orth}}$ in Eq. (16) require learning a scaled version of the CATE function up to an additive constant, with a scaling factor of $(1/\kappa)$. As the smoothness parameter $\kappa \to 0$, this scaling factor diverges and the minimizer becomes unbounded. At the same time, decreasing $\kappa$ also changes the *shape* of the loss around the optimum. The loss becomes increasingly flat along directions that preserve the treatment effect ordering: deviations of a candidate scoring function $g$ from the population minimizers that do not change the induced ranking are only weakly penalized. This behavior reflects that, in this limit, the orthogonal loss $\mathcal{L}^{\text{orth}}$ approaches the binary ranking loss $\mathcal{L}^{\text{bin}}$, which recovers only the ordering of treatment effects and does not admit a finite population minimizer (see Appendix D.2 for details).

Consequently, smaller values of $\kappa$ progressively reduce the learning objective from recovering the full shape of the CATE towards a pure ranking problem. As discussed above, decreasing $\kappa$ also amplifies the orthogonal correction for

---

[3]Note that $\mathcal{L}^{\text{orth}}$ and $\mathcal{L}^{\text{soft}}$ share the same set of population minimizers; for details see Appendix D.

| Learning objective | Population optimum | Constraint on $g$ (intuition) | Orthogonal |
|---|---|---|---|
| $\mathcal{L}^{\text{cate}}(g,\eta) = \mathbb{E}\big[(g(X) - \tau(X))^2\big]$ | $g(x) = \tau(x)$ | Identifies the full CATE function (learns magnitudes). | ✗ |
| $\mathcal{L}^{\text{bin}}(g,\eta) = \mathbb{E}\big[\ell(p_g(X, X'), b_\tau(X, X'))\big]$ | $g(x) = h\big(\tau(x)\big)$ with $h$ strictly increasing | Identifies the CATE ordering (learns ranking, not magnitudes). | ✗ |
| $\mathcal{L}^{\text{soft}}(g,\eta) = \mathbb{E}\big[\ell(p_g(X, X'), t_\tau(X, X'))\big]$ | | Identifies a scaled CATE up to a shift (interpolates between ranking and magnitudes via parameter $\kappa$). | ✗ |
| $\mathcal{L}^{\text{orth}}(g,\eta) = \mathbb{E}\big[\ell(p_g(X, X'), \tilde{t}_\eta(W, W'))\big]$ | $g(x) = \frac{1}{\kappa}\tau(x) + c$ | | ✓ |

*Table 2.* **Summary of learning objectives.** We compare four learning objectives, their population optima evaluated at the true nuisance functions, the induced constraints on the scoring function $g$, and whether they are Neyman-orthogonal. For $\mathcal{L}^{\text{bin}}$, the "optimum" denotes the class of order-preserving scoring functions that can approach the infimum arbitrarily closely (not attainable by any finite $g$). Our proposed orthogonal objective is highlighted .

pairs with similar treatment effects, which increases the variability of the pseudo labels in finite samples. The choice of $\kappa$ therefore reflects a bias-variance trade-off; we propose a practical strategy for selecting the value in the next section.

## 6. *Rank-Learner*: A two-stage learner for ranking treatment effects

In this section, we introduce *Rank-Learner*, an orthogonal two-stage learner for ranking individuals by their treatment effects from observational data. *Rank-Learner* proceeds as follows. ① In the first stage, we estimate nuisance functions $\hat{\eta}$ via cross-fitting using flexible machine learning models. ② In the second stage, we fit a scoring function $g$ by minimizing the empirical orthogonal ranking objective $\mathcal{L}^{\text{orth}}(g, \hat{\eta})$ on a random subsample of the training pairs. ③ At inference time, the scoring function $\hat{g}$ can be used directly to rank individuals, without having to perform any pairwise comparisons. An overview of our *Rank-Learner* is shown in Figure 2; we now describe each stage in detail.

① **Nuisance estimation.** In the first stage, we estimate the nuisance functions $\hat{\eta} = (\hat{\mu}_1, \hat{\mu}_0, \hat{e})$, with $\mu_t(x)$ the response surfaces and $e(x)$ the propensity score, using flexible machine learning models. Following standard practice in orthogonal learning, we use cross-fitted estimates $\hat{\eta}$ in the second-stage (Chernozhukov et al., 2018).

② **Orthogonal learning.** In the second stage, we train a scoring function $g$ by minimizing the empirical objective

$$\mathcal{L}^{\text{orth}}(g, \hat{\eta}) = \frac{1}{|\mathcal{P}|} \sum_{(i,j)\in\mathcal{P}} \ell\Big(p_g(x_i, x_j), \tilde{t}_{\hat{\eta}}(w_i, w_j)\Big) \quad (17)$$

where $\mathcal{P}$ is the set of sampled training pairs. For each pair $(i, j) \in \mathcal{P}$, the pairwise prediction of the model is $p_g(x_i, x_j)$ and the corresponding pseudo label is $\tilde{t}_{\hat{\eta}}(w_i, w_j)$. Importantly, orthogonality is incorporated *entirely* through these pseudo labels, while the loss itself retains the standard binary cross-entropy form. Consequently, the second-stage learning problem remains identical to standard pairwise

ranking, up to the construction of the training labels.

Since a dataset of size $n$ induces $|\mathcal{P}_{\text{all}}| = n^2$ possible training pairs, the second-stage optimization can become prohibitively expensive. To scale *Rank-Learner*, we therefore propose to optimize the loss using only a random subsample of pairs $\mathcal{P} \subset \mathcal{P}_{\text{all}}$ in each epoch, drawn uniformly from all available pairs. In our experiments, we show that ranking performance saturates quickly as the number of sampled pairs increases, suggesting that only a small fraction of pairs is usually sufficient.

We select $\kappa$ by maximizing an out-of-sample ranking criterion on a validation set. Concretely, we use the *area under the targeting operator curve* (AUTOC), which measures the average benefit of treating units in the order induced by a score (here $\hat{g}$), averaged across all treatment fractions (Yadlowsky et al., 2025). Since the AUTOC depends on unobserved treatment effects, we rely on the approximation proposed by Chernozhukov et al. (2025). The remaining hyperparameters are tuned analogously (see Appendix E.4).

③ **Inference.** The output of the second stage is a fitted scoring function $\hat{g} : \mathcal{X} \to \mathbb{R}$. Given a target population $\{x_i\}_{i=1}^m$, we compute scores $\hat{g}(x_i)$ and rank individuals accordingly, with larger values indicating higher predicted priority (i.e., a larger treatment effect in the induced ordering). Pairwise comparisons are only needed during training, since inference requires only pointwise evaluation of $\hat{g}$. For implementation details, see Appendix E.4.

## 7. Experiments

We evaluate *Rank-Learner* on synthetic, semi-synthetic, and real-world benchmarks. Synthetic and semi-synthetic experiments provide access to ground-truth treatment effects and allow controlled evaluation of ranking quality, following standard practice in the causal inference literature (Kamran et al., 2024; Kennedy, 2023; Curth & van der Schaar, 2021; Shalit et al., 2017). We additionally evaluate on the real-world CRITEO uplift benchmark (Diemert et al., 2021),

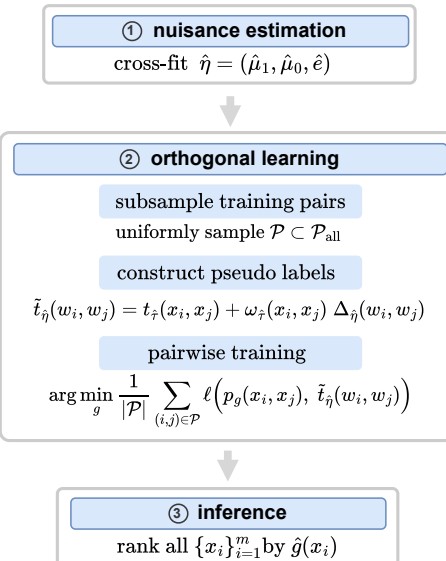

① **nuisance estimation**
cross-fit $\hat{\eta} = (\hat{\mu}_1, \hat{\mu}_0, \hat{e})$

② **orthogonal learning**

subsample training pairs
uniformly sample $\mathcal{P} \subset \mathcal{P}_{\text{all}}$

construct pseudo labels
$\tilde{t}_{\hat{\eta}}(w_i, w_j) = t_{\hat{\tau}}(x_i, x_j) + \omega_{\hat{\tau}}(x_i, x_j) \, \Delta_{\hat{\eta}}(w_i, w_j)$

pairwise training
$\arg\min_g \dfrac{1}{|\mathcal{P}|} \sum_{(i,j) \in \mathcal{P}} \ell\Big(p_g(x_i, x_j), \, \tilde{t}_{\hat{\eta}}(w_i, w_j)\Big)$

③ **inference**
rank all $\{x_i\}_{i=1}^m$ by $\hat{g}(x_i)$

*Figure 2.* **Overview of *Rank-Learner*.** In the first stage, we estimate nuisance functions (response surfaces and propensity score) via cross-fitting. In the second stage, we subsample training pairs, construct the pseudo labels, and learn a scoring function by minimizing the orthogonal ranking objective. Both stages can be instantiated with arbitrary machine learning models. At inference time, we rank individuals directly by their learned scores $\hat{g}(x)$.

where the training data is intentionally confounded while evaluation is performed on randomized test data. Full experimental details and extended results are provided in Appendix E and F.

### 7.1. Experimental design

**Setup.** In the synthetic and semi-synthetic benchmarks, we generate treatment effects as a strictly increasing, non-linear transformation of a latent score $s(x)$, such that the ground-truth ranking is fully determined by $s(x)$, while the treatment effect magnitudes depend on the chosen transformation. In the semi-synthetic setting, we use real covariates from established datasets covering distinct application domains – MOVIELENS (recommender systems), MIMIC-III (healthcare), and the CURRENT POPULATION SURVEY (public policy) – but construct treatments and outcomes following the same design principles as in the synthetic benchmark (Harper & Konstan, 2015; Johnson et al., 2016; Flood et al., 2025). For further details, we refer to Appendix E.1 and E.2.

Across the (semi-)synthetic experiments, we evaluate on a fixed test set of 1,000 samples and report results over five seeds. On the synthetic benchmark, we vary the training size from $n = 100$ to $n = 2,000$ to study ranking performance as a function of data availability. To separate nuisance estimation and second-stage learning, we use sample splitting:

one sample of size $n$ is used to fit the nuisance functions, and an independent sample of the same size $n$ is used to train the second-stage models (each with an internal train/validation split). In the semi-synthetic setting, we fix the training size to $n = 1,000$ and follow the same protocol. Tables report mean $\pm$ standard deviation over seeds, while figures show mean $\pm$ standard error.

**Baselines and training details.** We compare *Rank-Learner* against the naïve CATE-based strategy using the **T-learner** (Künzel et al., 2019) and the **DR-learner** (Kennedy, 2023), with the latter being Neyman-orthogonal. To isolate the benefit of orthogonalization for ranking, we also include a non-orthogonal **plug-in ranker** that learns a scoring model $g$ by optimizing $\mathcal{L}^{\text{soft}}(g, \hat{\eta})$. This learning objective uses the soft targets, based on plug-in estimates of the treatment effects, instead of the orthogonal pseudo labels. Together, these baselines allow us to disentangle *(i)* the benefit of direct ranking versus the strategy based on *full* CATE estimation and *(ii)* orthogonal versus plug-in objectives for ranking. Additional comparisons against recent CATE baselines are provided in Appendix F.4.

Across our experiments, we use the *same* model architecture and training details for a *fair* comparison. In particular, the nuisance functions and second-stage models are implemented as feedforward neural networks with a single hidden layer and ReLU activations, with linear output layers for regression and sigmoid output layers for classification. Models are trained with Adam (Kingma & Ba, 2015) for up to 50 epochs, with early stopping based on the validation loss, retaining the best model checkpoint. For model selection, we tune the hyperparameters of the ranking methods using the approximated AUTOC on the validation set (Chernozhukov et al., 2025), while CATE estimators are selected using their standard validation loss. Training is computationally lightweight with all models converging within minutes. Further implementation details are in Appendix E.4.

**Evaluation metrics.** Our primary evaluation metric for the (semi-)synthetic benchmarks is the AUTOC (Yadlowsky et al., 2025). The AUTOC evaluates how well a method ranks individuals by their treatment effects, by averaging the cumulative treatment benefit among the top-ranked individuals over all treatment fractions. In our evaluation, we compute AUTOC on the test set using the ground-truth treatment effects. Unlike AUROC, AUTOC is not normalized and therefore not constrained to $[0, 1]$. For the real-world uplift benchmark, ground-truth treatment effects are unknown, and we instead evaluate ranking quality using AUUC. For completeness, we also report the *mean policy value* in Appendix F.1, which evaluates the quality of the implied policies.

*Table 3.* **Synthetic benchmark (main results).** Test AUTOC (mean $\pm$ std dev over five seeds) across training sizes. *Higher is better.* The oracle column reports AUTOC obtained by ranking the test set using the true treatment effects. Best mean is shown in **bold**.

| Method | $n = 100$ | $n = 250$ | $n = 500$ | $n = 1,000$ | $n = 2,000$ | oracle |
|---|---|---|---|---|---|---|
| T-learner | $0.88 \pm 0.17$ | $0.96 \pm 0.14$ | $1.24 \pm 0.05$ | $1.32 \pm 0.02$ | $1.36 \pm 0.00$ | |
| DR-learner | $0.80 \pm 0.18$ | $1.16 \pm 0.12$ | $1.28 \pm 0.05$ | $1.33 \pm 0.02$ | $1.36 \pm 0.02$ | 1.40 |
| Plug-in ranker | $0.69 \pm 0.32$ | $0.95 \pm 0.14$ | $1.24 \pm 0.06$ | $1.31 \pm 0.02$ | $1.36 \pm 0.00$ | |
| Rank-learner (*ours*) | $\mathbf{1.00 \pm 0.19}$ | $\mathbf{1.28 \pm 0.03}$ | $\mathbf{1.31 \pm 0.01}$ | $\mathbf{1.34 \pm 0.01}$ | $\mathbf{1.37 \pm 0.00}$ | |

## 7.2. Synthetic benchmark results

Table 3 reports test AUTOC across training sizes on the synthetic benchmark. We draw three main findings. **(1)** *Rank-Learner* consistently outperforms standard pointwise CATE estimators (T- and DR-learners) across all sample sizes, demonstrating the benefit of directly targeting treatment effect ranking rather than recovering effect magnitudes. **(2)** Compared to the non-orthogonal plug-in ranker, *Rank-Learner* achieves systematically higher AUTOC, with the largest improvements observed in small-sample regimes where nuisance estimation error is most pronounced. This highlights the advantage of orthogonalization for ranking. **(3)** As training size increases, performance differences across methods become smaller, reflecting improved nuisance estimation quality in our controlled setting (see Appendix F.1, Table 7). Importantly, the relative ordering of methods remains unchanged, with *Rank-Learner* achieving the strongest performance throughout. Taken together, these results show that, among the considered baselines, directly targeting the ranking task with an orthogonal learning objective yields the best performance.

**Pair subsampling.** Figure 3 shows the computational trade-off from subsampling training pairs in the second-stage ranker, and its effect on ranking performance. *Rank-Learner* attains strong performance in terms of AUTOC, even when using only a small fraction of pairs per epoch. This suggests that the second-stage performance is mostly driven by the quality of the nuisance estimates, and that subsampling is an effective way to scale our *Rank-Learner*. Additional analyses, including an alternative pair sampling strategy, are provided in Appendix F.2.

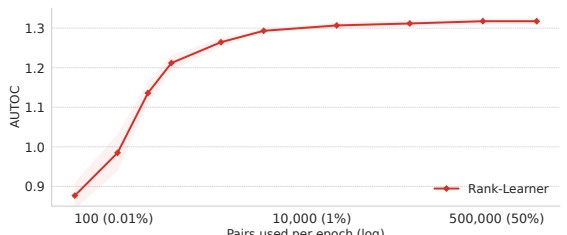

*Figure 3.* **Synthetic benchmark (pair subsampling).** Test AUTOC (mean $\pm$ s.e. over five seeds) of *Rank-Learner* as a function of the number of sampled training pairs per epoch ($n = 1,000$ with $n^2 = 10^6$ possible training pairs). *Higher is better.* The horizontal axis shows the fraction of pairs used per epoch (log).

**Sensitivity to overlap.** Figure 4 studies the robustness to limited overlap by varying the propensity mechanism while keeping the remaining components of the synthetic data-generating process fixed (see Appendix F.3 for details). As overlap decreases, ranking performance deteriorates for all methods, which is expected and reflects the increasing difficulty of causal inference when treated and control groups become less comparable. Importantly, across the full range of overlap levels considered, our *Rank-Learner* consistently achieves the highest mean AUTOC. Additional sensitivity analyses, including robustness to the choice of $\kappa$, nuisance misspecification, and pseudo label clipping, are provided in Appendix F.3.

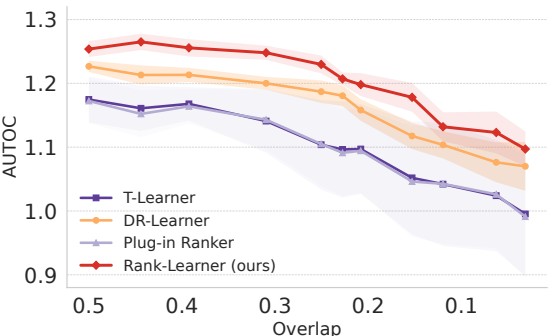

*Figure 4.* **Synthetic benchmark (overlap sensitivity).** Test AUTOC (mean $\pm$ s.e. over five seeds) as a function of overlap (decreasing left to right) for $n = 500$. *Higher is better.* Overlap is varied by changing treatment assignment, remaining components of the synthetic data-generating process are kept fixed.

## 7.3. Semi-synthetic benchmark results

We next evaluate *Rank-Learner* on the three semi-synthetic benchmarks with realistic covariate distributions (MOVIELENS, MIMIC-III, and CPS). This experiment tests whether the benefits of orthogonal ranking extend beyond synthetic data. Table 4 reports test AUTOC at a fixed training size of $n = 1,000$. At this sample size, nuisance estimation error remains considerable, mirroring the difficulty of learning nuisance functions in observational settings. Across all datasets, *Rank-Learner* achieves the strongest ranking performance among the considered baselines, with consistent improvements over the plug-in ranker and the pointwise CATE estimators.

*Table 4.* **Semi-synthetic benchmarks**. Test AUTOC (mean $\pm$ std dev over five seeds) with training size $n = 1,000$. *Higher is better.* The oracle row reports AUTOC obtained by ranking the test set using the true treatment effects. Best mean is shown in **bold**.

| Method | MOVIELENS | MIMIC-III | CPS |
|---|---|---|---|
| oracle | 1.39 | 1.22 | 1.01 |
| T-learner | $1.31 \pm 0.03$ | $1.12 \pm 0.05$ | $0.87 \pm 0.08$ |
| DR-learner | $1.34 \pm 0.02$ | $1.17 \pm 0.02$ | $0.92 \pm 0.02$ |
| Plug-in ranker | $1.30 \pm 0.03$ | $1.11 \pm 0.05$ | $0.87 \pm 0.08$ |
| Rank-learner (*ours*) | $\mathbf{1.35 \pm 0.01}$ | $\mathbf{1.18 \pm 0.02}$ | $\mathbf{0.95 \pm 0.01}$ |

### 7.4. Real-world uplift benchmark results

We finally evaluate *Rank-Learner* on the real-world CRITEO uplift benchmark (Diemert et al., 2021), which contains randomized data from online advertising campaigns. Each observation consists of user covariates, a treatment indicator corresponding to ad exposure, and a binary outcome indicating whether the user visited the advertiser's website.

To evaluate *Rank-Learner* on CRITEO in an observational setting, we follow Diemert et al. (2021) and induce confounding in the training data by selectively sampling training instances, while leaving the test data randomized. We use sample splitting with fixed train and validation sizes of 10,000 and 5,000 samples, respectively. Since ground-truth treatment effects are unknown, we evaluate ranking quality using *area under the uplift curve* (AUUC) on the test data. Due to the severe treatment imbalance and rare outcomes in this challenging benchmark, we report results on test sets of increasing size to reduce evaluation noise. Additional details on the experimental design, confounding mechanism, and evaluation protocol are provided in Appendix E.3.

Table 5 reports AUUC on the randomized test data. Across all test sizes, *Rank-Learner* achieves the strongest mean AUUC among the considered baselines. Overall, these results further support the benefit of directly optimizing an orthogonal ranking objective in observational settings.

*Table 5.* **Criteo uplift benchmark.** AUUC on randomized test sets of increasing size (50k, 500k, and 1M). Results are reported as mean $\pm$ std dev over five seeds. Values are reported $\times 10^3$ for readability. *Higher is better.* Best mean is shown in **bold**.

| Method | AUUC $\times 10^3$ | | |
|---|---|---|---|
| | 50k | 500k | 1M |
| T-learner | $3.74 \pm 1.18$ | $5.09 \pm 1.59$ | $5.08 \pm 1.62$ |
| DR-learner | $4.44 \pm 1.12$ | $5.01 \pm 1.04$ | $5.17 \pm 1.13$ |
| Plug-in ranker | $3.78 \pm 1.59$ | $4.99 \pm 1.69$ | $5.04 \pm 1.65$ |
| Rank-learner (*ours*) | $\mathbf{5.19 \pm 1.87}$ | $\mathbf{5.83 \pm 0.57}$ | $\mathbf{5.90 \pm 0.40}$ |

## 8. Conclusion

In this paper, we introduced *Rank-Learner*, the first Neyman-orthogonal two-stage learner for ranking individuals by their treatment effects from observational data. Many downstream decision-making tasks only require recovering the relative ordering of treatment effects, rather than accurately estimating their exact magnitudes. To this end, we derived a novel orthogonal ranking objective that is robust to nuisance estimation errors. In addition, *Rank-Learner* is model-agnostic and can be instantiated with arbitrary machine learning models. We demonstrated the strong ranking performance of *Rank-Learner* across a broad range of experiments on synthetic, semi-synthetic, and real-world benchmarks.

## Impact statement

This paper presents methodological work aimed at advancing the field of machine learning and causal inference. While ranking individuals by treatment effects may inform decision-making in domains such as healthcare, marketing, or public policy, the proposed method is not tied to any specific application. As with all causal methods based on observational data, responsible use requires careful consideration of the underlying assumptions (e.g., unconfoundedness) and the context in which the estimates / rankings are applied.

## Acknowledgments

This paper is supported by the DAAD program Konrad Zuse Schools of Excellence in Artificial Intelligence, sponsored by the Federal Ministry of Research, Technology and Space. This research also received funding from the Research Foundation Flanders (FWO Vlaanderen) with grant number 11Q2C24N and from the Flemish government under the "Onderzoeksprogramma Artificiële Intelligentie (AI) Vlaanderen" program.

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

# A. Extended literature review

## A.1. Policy learning

In policy learning, the goal is to learn a treatment assignment rule $\pi : \mathcal{X} \to \{0, 1\}$ from observational data that maximizes the so-called *policy value* $V(\pi) = \mathbb{E}[Y(\pi(X))]$, which represents the expected outcome in the population under policy $\pi(x)$. There has been extensive work on this topic (Qian & Murphy, 2011), including methods designed for limited overlap (Kallus, 2021; 2018), unobserved confounding (Hess et al., 2026), doubly-robust methods (Athey & Wager, 2021), extensions to continuous treatments (Schweisthal et al., 2023), fairness-constrained policies (Frauen et al., 2024), and interpretable CATE-based policies (Frauen et al., 2025).

Many treatment assignment problems involve budget or capacity constraints, where only a subset of individuals can be treated. In such settings, learning a ranking of individuals according to their treatment effects is often more useful than learning a single binary treatment policy tailored to a fixed constraint. While policy learning methods *can* optimize treatment rules for a specific budget or constrained policy class, a learned ranking under *Rank-Learner* can instead be used downstream to instantiate different treatment policies for different treatment budgets, without having to retrain the model for each choice.

A different class of constraints arises when treatment assignment depends on whether treatment effects exceed a prespecified threshold. For such threshold-based treatment decisions, preserving only the relative ordering of treatment effects is not sufficient, since decisions additionally depend on magnitude information near the decision boundary. Recent work has therefore studied methods specifically designed for such threshold-based treatment decisions (e.g. Frauen et al., 2025).

## A.2. Uplift modeling

Our work is also closely related to uplift modeling (Diemert et al., 2021; Devriendt et al., 2021), which studies how to model the incremental impact of an intervention on individual outcomes, often in applications such as marketing or recommender systems. Classical uplift modeling typically focuses on binary treatments and outcomes, where the goal is to identify responders and non-responders to treatment. More recent uplift modeling work has also considered settings with continuous outcomes and multi-treatment settings (Verbeken et al., 2025; Olaya et al., 2020). In this sense, *Rank-Learner* naturally connects to the broader uplift modeling literature, as illustrated by our experiments on the CRITEO uplift benchmark. Our contribution differs in that we directly study treatment effect ranking from observational data and propose the first Neyman-orthogonal two-stage learner for this task.

# B. Orthogonal learning based on influence functions

In this appendix, we provide a brief background on influence functions and how they can be used to construct orthogonal losses following Kennedy (2024).

**Influence functions.** We consider a setting where we observe data $W = (X, T, Y)$ distributed according to an unknown probability distribution $\mathbb{P}$ that lies in some statistical model $\mathcal{P}$ (a set of distributions). The goal is to estimate a statistical quantity of interest that can be expressed as a functional $\psi : \mathcal{P} \to \mathbb{R}$ (as an example, consider the average treatment effect $\psi(\mathbb{P}) = \mathbb{E}_X \left[ \mathbb{E}[Y \mid T = 1, X] - \mathbb{E}[Y \mid T = 0, X] \right]$). Intuitively, the influence function $\mathbb{IF}_\psi(w, \mathbb{P})$ captures the sensitivity of the target functional to small perturbations of the distribution $\mathbb{P}$, obtained by adding an infinitesimal amount of probability mass on the observation $W = w$.

**Plug-in bias and the one-step correction.** A natural way to estimate the functional of interest is to directly replace the true distribution $\mathbb{P}$ with a finite-sample estimate $\hat{\mathbb{P}}$ to obtain the *plug-in* estimator $\psi(\hat{\mathbb{P}})$. However, since $\hat{\mathbb{P}}$ only approximates $\mathbb{P}$, plug-in estimators typically suffer from a first-order bias term that can be expressed through the influence function

$$\psi(\hat{\mathbb{P}}) - \psi(\mathbb{P}) = -\int \mathbb{IF}_\psi(w, \hat{\mathbb{P}}) \mathrm{d}\mathbb{P}(w) + R_2, \tag{18}$$

where $R_2$ collects higher-order remainder terms that are negligible under mild conditions. This implies that simply "plugging-in" an estimate of the distribution $\mathbb{P}$ into the functional $\psi$ yields a biased estimator. In order to correct this bias, the term involving the influence function can be estimated from the sample and added back to the plug-in estimator, resulting in the one-step corrected estimator

$$\hat{\psi}^{\text{one-step}} = \psi(\hat{\mathbb{P}}) + \mathbb{P}_n \left[ \mathbb{IF}_\psi(W, \hat{\mathbb{P}}) \right]. \tag{19}$$

**Orthogonal loss construction.** In this work, the goal is *not* to estimate a finite-dimensional target parameter such as the average treatment effect. Instead, we aim to learn a model $g(x)$ that preserves the ranking induced by the conditional average treatment effect $\tau(x)$, which is an infinite-dimensional quantity. Therefore, the one-step correction described above is not directly applicable to our setting. However, the population loss $\mathcal{L}(g, \eta)$ that is used to learn $g$ depends on unknown nuisance components $\eta = (\mu_1, \mu_0, e)$, which each depend on $\mathbb{P}$. In practice, these components must be replaced by estimates $\mathcal{L}(g, \hat{\eta})$, which introduce the same kind of *plug-in bias* as in the standard setting with a finite-dimensional target quantity. By applying the correction based on the influence function directly to the loss, we obtain an orthogonal loss whose gradients are first-order insensitive to estimation errors in the nuisances (i.e., Neyman-orthogonality; see Section 4 for a background).

# C. Derivation and proof of the Neyman-orthogonal ranking loss

In Appendix B, we gave the intuition behind orthogonal learning based on influence functions and why it is desirable in our setting. In this appendix, we formally derive our proposed loss and prove that it is Neyman-orthogonal.

## C.1. Derivation of the efficient influence function

**Strategy.** In order to derive the efficient influence function of our loss, we follow the strategy proposed by Kennedy (2024) and Hines et al. (2022). We work with a nonparametric statistical model $\mathcal{P}$, defined as the set of all probability distributions for the observed data $W = (X, T, Y)$. Let $\mathbb{P} \in \mathcal{P}$ denote the true data-generating distribution of $W$ and consider the one-dimensional parametric submodel

$$\mathcal{P}_\epsilon = \left\{ \mathbb{P}_\epsilon = (1 - \epsilon)\mathbb{P} + \epsilon \delta_w \mid \epsilon \in [0, 1) \right\}, \qquad \text{with} \quad \mathcal{P}_\epsilon \subseteq \mathcal{P}, \tag{20}$$

where $\delta_w$ denotes a point-mass distribution that assigns all its probability mass to the observation $W = w$. For this choice of submodel, the influence function of any functional $\psi(\mathbb{P})$ is given by the directional derivative

$$\mathbb{IF}_\psi(w, \mathbb{P}) = \left. \frac{\mathrm{d}}{\mathrm{d}\epsilon} \psi(\mathbb{P}_\epsilon) \right|_{\epsilon=0}. \tag{21}$$

This characterization allows us to derive the efficient influence function of our loss by differentiating $\psi(\mathbb{P}_\epsilon) = \mathcal{L}(g, \eta_\epsilon)$ with respect to $\epsilon$. This can be carried out by using standard differentiation rules and by treating the expectations inside $\mathcal{L}(g, \eta_\epsilon)$ as finite sums, as if the data were discrete. For a comprehensive background and technical details, we refer to (Kennedy, 2024) and (Hines et al., 2022).

**Derivation.** Recall that our goal is to learn a scoring function $g(x)$ that preserves the ordering induced by the conditional average treatment effect $\tau(x)$. To this end, we consider the pairwise population loss

$$\mathcal{L}^{\text{soft}}(g, \eta) = \mathbb{E}_{X,X'}\left[ \ell\big(p_g(X, X'),\, t_\tau(X, X')\big) \right] \tag{22}$$

$$= \mathbb{E}_{X,X'}\left[ -t_\tau(X, X') \log p_g(X, X') - \big(1 - t_\tau(X, X')\big) \log\big(1 - p_g(X, X')\big) \right], \tag{23}$$

where $\ell(p, t) = -t \log p - (1 - t) \log(1 - p)$ denotes the binary cross-entropy loss. For brevity, we sometimes write $\ell(X, X')$ to denote $\ell\big(p_g(X, X'),\, t_\tau(X, X')\big)$, where we use

$$p_g(X, X') = \sigma\big(g(X) - g(X')\big), \qquad \text{and} \quad t_\tau(X, X') = \sigma\left( \frac{\tau(X) - \tau(X')}{\kappa} \right). \tag{24}$$

Here, $\kappa > 0$ is a smoothness parameter, and the nuisance components $\eta$ enter the loss through $\tau(X) = \mu_1(X) - \mu_0(X)$.

The influence function of this loss evaluated at $w_0 = (x_0, t_0, y_0)$ can be written based on the product rule as

$$\mathbb{IF}_{\mathcal{L}^{\text{soft}}}(w_0, \mathbb{P}) = \underbrace{\sum_x \sum_{x'} \mathbb{IF}_{\ell(x,x')}(w_0, \mathbb{P}) p(x) p(x')}_{A(w_0, \mathbb{P})} + \underbrace{\sum_x \sum_{x'} \ell(x, x') \mathbb{IF}_{p(x)p(x')}(w_0, \mathbb{P})}_{B(w_0, \mathbb{P})}. \tag{25}$$

Considering the $A(w_0, \mathbb{P})$ term, for fixed $(x, x')$, the dependence of $\ell(x, x')$ on $\mathbb{P}$ is only through $t_\tau(x, x')$, and by the chain rule we have

$$\mathbb{IF}_{\ell(x,x')}(w_0, \mathbb{P}) = \frac{\partial \ell(x, x')}{\partial t_\tau(x, x')} \frac{\partial t_\tau(x, x')}{\partial \big(\tau(x) - \tau(x')\big)} \mathbb{IF}_{\tau(x)-\tau(x')}(w_0, \mathbb{P}) \tag{26}$$

$$= \underbrace{\log\left( \frac{1 - p_g(x, x')}{p_g(x, x')} \right) \left( \frac{t_\tau(x, x')\big(1 - t_\tau(x, x')\big)}{\kappa} \right)}_{C(x, x')} \left( \mathbb{IF}_{\tau(x)}(w_0, \mathbb{P}) - \mathbb{IF}_{\tau(x')}(w_0, \mathbb{P}) \right) \tag{27}$$

$$= C(x, x') \left( \mathbb{IF}_{\tau(x)}(w_0, \mathbb{P}) - \mathbb{IF}_{\tau(x')}(w_0, \mathbb{P}) \right). \tag{28}$$

To evaluate the term $\mathbb{IF}_{\tau(x)}(w_0, \mathbb{P})$, we use the known influence function of the CATE at a fixed covariate value $x$, given by

$$\mathbb{IF}_{\tau(x)}(w_0, \mathbb{P}) = \frac{\mathbf{I}\{x_0 = x\}}{p(x)}\left(\frac{t_0}{e(x)}\big(y_0 - \mu_1(x)\big) - \frac{1 - t_0}{1 - e(x)}\big(y_0 - \mu_0(x)\big)\right) \tag{29}$$

$$= \frac{\mathbf{I}\{x_0 = x\}}{p(x)}\big(\phi_\eta(w_0, x) - \tau(x)\big), \tag{30}$$

where the last line is obtained by adding and subtracting $\tau(x)$, and denoting $\phi_\eta(w_0, x)$ the doubly robust score for $\tau(x)$, which depends on the nuisances $\eta = (\mu_1, \mu_0, e)$. Plugging this into the expression for $A(w_0, \mathbb{P})$, we get

$$A(w_0, \mathbb{P}) = \sum_x \sum_{x'} C(x, x')\,\mathbf{I}\{x_0 = x\}\big(\phi_\eta(w_0, x) - \tau(x)\big)\,p(x')$$

$$- \sum_x \sum_{x'} C(x, x')\,\mathbf{I}\{x_0 = x'\}\big(\phi_\eta(w_0, x') - \tau(x')\big)\,p(x) \tag{31}$$

$$= \sum_{x'} C(x_0, x')\big(\phi_\eta(w_0, x_0) - \tau(x_0)\big)\,p(x') - \sum_x C(x, x_0)\big(\phi_\eta(w_0, x_0) - \tau(x_0)\big)\,p(x) \tag{32}$$

$$= \mathbb{E}_{X'}\big[C(x_0, X')\big]\big(\phi_\eta(w_0) - \tau(x_0)\big) - \mathbb{E}_X\big[C(X, x_0)\big]\big(\phi_\eta(w_0) - \tau(x_0)\big), \tag{33}$$

where for brevity we write $\phi_\eta(w_0)$ rather than $\phi_\eta(w_0, x_0)$, since $w_0 = (x_0, t_0, y_0)$ already contains the covariates $x_0$ on which the score depends.

Now consider the $B(w_0, \mathbb{P})$ term, where, by the product rule, we have

$$B(w_0, \mathbb{P}) = \sum_x \sum_{x'} \ell(x, x')\mathbb{IF}_{p(x)p(x')}(w_0, \mathbb{P}) \tag{34}$$

$$= \sum_x \sum_{x'} \ell(x, x')\,\mathbb{IF}_{p(x)}(w_0, \mathbb{P})\,p(x') + \sum_x \sum_{x'} \ell(x, x')\,p(x)\,\mathbb{IF}_{p(x')}(w_0, \mathbb{P}). \tag{35}$$

To evaluate the term $\mathbb{IF}_{p(x)}(w_0, \mathbb{P})$, we use the known influence function of a marginal distribution at a fixed covariate value $x$, given by

$$\mathbb{IF}_{p(x)}(w_0, \mathbb{P}) = \mathbf{I}\{x_0 = x\} - p(x). \tag{36}$$

Plugging this into the equation for $B(w_0, \mathbb{P})$ gives

$$B(w_0, \mathbb{P}) = \sum_x \sum_{x'} \ell(x, x')\,\mathbf{I}\{x_0 = x\}\,p(x') - \sum_x \sum_{x'} \ell(x, x')\,p(x)p(x') \tag{37}$$

$$+ \sum_x \sum_{x'} \ell(x, x')\,\mathbf{I}\{x_0 = x'\}\,p(x) - \sum_x \sum_{x'} \ell(x, x')\,p(x)p(x')$$

$$= \mathbb{E}_{X'}\big[\ell(x_0, X')\big] - \mathbb{E}_{X,X'}\big[\ell(X, X')\big] + \mathbb{E}_X\big[\ell(X, x_0)\big] - \mathbb{E}_{X,X'}\big[\ell(X, X')\big]. \tag{38}$$

Putting everything together, the efficient influence function of the population loss $\mathcal{L}^{\text{soft}}$ becomes

$$\mathbb{IF}_{\mathcal{L}^{\text{soft}}}(w_0, \mathbb{P}) = A(w_0, \mathbb{P}) + B(w_0, \mathbb{P}) \tag{39}$$

$$= \mathbb{E}_{X'}\big[C(x_0, X')\big]\big(\phi_\eta(w_0) - \tau(x_0)\big) - \mathbb{E}_X\big[C(X, x_0)\big]\big(\phi_\eta(w_0) - \tau(x_0)\big) \tag{40}$$

$$+ \mathbb{E}_{X'}\big[\ell(x_0, X')\big] - \mathbb{E}_{X,X'}\big[\ell(X, X')\big] + \mathbb{E}_X\big[\ell(X, x_0)\big] - \mathbb{E}_{X,X'}\big[\ell(X, X')\big],$$

with

$$C(x, x') = \log\left(\frac{1 - p_g(x, x')}{p_g(x, x')}\right)\left(\frac{t_\tau(x, x')\big(1 - t_\tau(x, x')\big)}{\kappa}\right). \tag{41}$$

### C.2. One-step correction

As explained in Appendix B, the one-step correction removes the first-order bias from the plug-in estimator by adding the empirical mean of the efficient influence function evaluated at the empirical distribution $\hat{\mathbb{P}}$. In our setting, the resulting estimator becomes

$$\mathcal{L}^{\text{orth}}(g, \hat{\eta}) = \mathcal{L}^{\text{soft}}(g, \hat{\eta}) + \mathbb{P}_n\left[\mathbb{IF}_{\mathcal{L}^{\text{soft}}}(W, \hat{\mathbb{P}})\right] \tag{42}$$

$$= \mathcal{L}^{\text{soft}}(g, \hat{\eta}) + \frac{1}{n}\sum_{i=1}^{n} \mathbb{IF}_{\mathcal{L}^{\text{soft}}}(w_i, \hat{\mathbb{P}}). \tag{43}$$

Since the influence function is evaluated at the empirical distribution $\hat{\mathbb{P}}$, all expectations are taken under $\hat{\mathbb{P}}$. Concretely, for any fixed $x_i$, we have

$$\hat{\mathbb{E}}_{X'}\left[C(x_i, X')\right] = \frac{1}{n}\sum_{j=1}^{n} C(x_i, x_j), \tag{44}$$

and, for the double expectation, we have

$$\hat{\mathbb{E}}_{X,X'}\left[\ell(X, X')\right] = \frac{1}{n^2}\sum_{j=1}^{n}\sum_{k=1}^{n} \ell(x_j, x_k). \tag{45}$$

We can now show that the empirical mean of the remainder term $B(W, \hat{\mathbb{P}})$ vanishes. Evaluating all expectations under the empirical distribution $\hat{\mathbb{P}}$, we have

$$\mathbb{P}_n\left[B(W, \hat{\mathbb{P}})\right] = \frac{1}{n}\sum_{i=1}^{n}\left\{\hat{\mathbb{E}}_{X'}\left[\ell(x_i, X')\right] - \hat{\mathbb{E}}_{X,X'}\left[\ell(X, X')\right] + \hat{\mathbb{E}}_X\left[\ell(X, x_i)\right] - \hat{\mathbb{E}}_{X,X'}\left[\ell(X, X')\right]\right\} \tag{46}$$

$$= \frac{1}{n}\sum_{i=1}^{n}\left\{\frac{1}{n}\sum_{j=1}^{n}\ell(x_i, x_j) - \frac{1}{n^2}\sum_{j=1}^{n}\sum_{k=1}^{n}\ell(x_j, x_k) + \frac{1}{n}\sum_{j=1}^{n}\ell(x_j, x_i) - \frac{1}{n^2}\sum_{j=1}^{n}\sum_{k=1}^{n}\ell(x_j, x_k)\right\} \tag{47}$$

$$= 0, \tag{48}$$

where the last equality follows by relabeling indices in the double sums.

Now consider the empirical mean of the $A(W, \hat{\mathbb{P}})$ term:

$$\mathbb{P}_n\left[\mathbb{IF}_{\mathcal{L}^{\text{soft}}}(W, \hat{\mathbb{P}})\right] = \mathbb{P}_n\left[A(W, \hat{\mathbb{P}})\right] \tag{49}$$

$$= \frac{1}{n}\sum_{i=1}^{n}\hat{\mathbb{E}}_{X'}\left[C(x_i, X')\right]\left(\phi_{\hat{\eta}}(w_i) - \hat{\tau}(x_i)\right) - \frac{1}{n}\sum_{i=1}^{n}\hat{\mathbb{E}}_X\left[C(X, x_i)\right]\left(\phi_{\hat{\eta}}(w_i) - \hat{\tau}(x_i)\right) \tag{50}$$

$$= \frac{1}{n^2}\sum_{i=1}^{n}\sum_{j=1}^{n}C(x_i, x_j)\left(\left(\phi_{\hat{\eta}}(w_i) - \hat{\tau}(x_i)\right) - \left(\phi_{\hat{\eta}}(w_j) - \hat{\tau}(x_j)\right)\right), \tag{51}$$

where the last equality again follows by relabeling indices. Combining the plug-in loss with the one-step correction gives the empirical loss

$$\mathcal{L}^{\text{orth}}(g, \hat{\eta}) = \mathcal{L}^{\text{soft}}(g, \hat{\eta}) + \mathbb{P}_n\left[\mathbb{IF}_{\mathcal{L}^{\text{soft}}}(W, \hat{\mathbb{P}})\right] \tag{52}$$

$$= \frac{1}{n^2}\sum_{i=1}^{n}\sum_{j=1}^{n}\left\{-t_{\hat{\tau}}(x_i, x_j)\log p_g(x_i, x_j) - \left(1 - t_{\hat{\tau}}(x_i, x_j)\right)\log\left(1 - p_g(x_i, x_j)\right)\right. \tag{53}$$

$$\left. + C(x_i, x_j)\left(\left(\phi_{\hat{\eta}}(w_i) - \hat{\tau}(x_i)\right) - \left(\phi_{\hat{\eta}}(w_j) - \hat{\tau}(x_j)\right)\right)\right\}. \tag{54}$$

Recall that the correction weight satisfies

$$C(x_i, x_j) = \log\left(\frac{1 - p_g(x_i, x_j)}{p_g(x_i, x_j)}\right)\left(\frac{t_{\hat{\tau}}(x_i, x_j)\big(1 - t_{\hat{\tau}}(x_i, x_j)\big)}{\kappa}\right), \tag{55}$$

where the first factor is exactly the derivative of the binary cross-entropy loss with respect to its label

$$\frac{\partial}{\partial t}\Big(-t\log p - (1-t)\log(1-p)\Big) = \log\left(\frac{1-p}{p}\right). \tag{56}$$

Since binary cross-entropy is affine in the label $t$, multiplying this derivative by any quantity can be absorbed into the label itself. Therefore, the entire correction is equivalent to using standard binary cross-entropy loss with pseudo labels

$$\tilde{t}_{\hat{\eta}}(w_i, w_j) = t_{\hat{\tau}}(x_i, x_j) + \frac{t_{\hat{\tau}}(x_i, x_j)\big(1 - t_{\hat{\tau}}(x_i, x_j)\big)}{\kappa}\Big(\big(\phi_{\hat{\eta}}(w_i) - \hat{\tau}(x_i)\big) - \big(\phi_{\hat{\eta}}(w_j) - \hat{\tau}(x_j)\big)\Big), \tag{57}$$

so that

$$\mathcal{L}^{\text{orth}}(g, \hat{\eta}) = \frac{1}{n^2}\sum_{i=1}^{n}\sum_{j=1}^{n}\Big(-\tilde{t}_{\hat{\eta}}(w_i, w_j)\log p_g(x_i, x_j) - \big(1 - \tilde{t}_{\hat{\eta}}(w_i, w_j)\big)\log\big(1 - p_g(x_i, x_j)\big)\Big). \tag{58}$$

### C.3. Population loss and pseudo labels

At the population level, the derived loss can be written as binary cross-entropy between the model predictions and the pseudo labels

$$\mathcal{L}^{\text{orth}}(g, \eta) = \mathbb{E}_{W, W'}\Big[-\tilde{t}_{\eta}(W, W')\log p_g(X, X') - \big(1 - \tilde{t}_{\eta}(W, W')\big)\log\big(1 - p_g(X, X')\big)\Big], \tag{59}$$

with

$$\tilde{t}_{\eta}(W, W') = t_{\tau}(X, X') + \frac{t_{\tau}(X, X')\big(1 - t_{\tau}(X, X')\big)}{\kappa}\Big(\big(\phi_{\eta}(W) - \tau(X)\big) - \big(\phi_{\eta}(W') - \tau(X')\big)\Big). \tag{60}$$

### C.4. Proof of Neyman-orthogonality

*Proof of Theorem 5.1.* Let $\eta^0 = (\mu_1^0, \mu_0^0, e^0)$ denote the true nuisance components, and let $g^0$ be a population minimizer of the loss $\mathcal{L}^{\text{orth}}(g, \eta^0)$. Neyman-orthogonality requires that for any perturbation direction $\Delta g$ and $\Delta \eta$,

$$D_{\eta}D_g\,\mathcal{L}^{\text{orth}}(g^0, \eta^0)[\Delta g, \Delta \eta] = 0. \tag{61}$$

We verify this condition by explicitly computing the cross-derivative evaluated at $(g^0, \eta^0)$ and showing that it equals zero for each nuisance component, following the approach of Morzywolek et al. (2024).

**Directional derivative with respect to $g$.** We first write the loss in binary cross-entropy form

$$\mathcal{L}^{\text{orth}}(g, \eta) = \mathbb{E}_{W, W'}\big[\ell_{g, \eta}(W, W')\big] \tag{62}$$

$$= \mathbb{E}_{W, W'}\Big[-\tilde{t}_{\eta}(W, W')\log p_g(X, X') - \big(1 - \tilde{t}_{\eta}(W, W')\big)\log\big(1 - p_g(X, X')\big)\Big], \tag{63}$$

where $p_g(X, X') = \sigma\big(g(X) - g(X')\big)$ and $\tilde{t}_{\eta}(W, W')$ are the pseudo labels defined in Eq. (60).

To compute the directional derivative with respect to $g$, we consider the perturbation path

$$g_t(x) = g^0(x) + t\Delta g(x), \qquad \text{with} \quad \Delta g(x) = g(x) - g^0(x), \tag{64}$$

such that, under mild regularity conditions,

$$D_g\mathcal{L}^{\text{orth}}(g^0, \eta)[\Delta g] = \frac{\mathrm{d}}{\mathrm{d}t}\mathcal{L}^{\text{orth}}(g_t, \eta)\Big|_{t=0} = \mathbb{E}_{W, W'}\left[\frac{\mathrm{d}}{\mathrm{d}t}\ell_{g_t, \eta}(W, W')\Big|_{t=0}\right]. \tag{65}$$

To evaluate the inner derivative, we apply the chain rule

$$
\frac{\mathrm{d}}{\mathrm{d}t}\,\ell_{g_t,\eta}(W,W')\bigg|_{t=0} = \frac{\partial \ell_{g_t,\eta}(W,W')}{\partial p_{g_t}(X,X')}\bigg|_{t=0} \frac{\partial p_{g_t}(X,X')}{\partial\big(g_t(X) - g_t(X')\big)}\bigg|_{t=0} \frac{\partial\big(g_t(X) - g_t(X')\big)}{\partial t}\bigg|_{t=0} \tag{66}
$$

$$
= \frac{p_{g^0}(X,X') - \tilde{t}_\eta(W,W')}{p_{g^0}(X,X')\big(1 - p_{g^0}(X,X')\big)}\,p_{g^0}(X,X')\big(1 - p_{g^0}(X,X')\big)\big(\Delta g(X) - \Delta g(X')\big) \tag{67}
$$

$$
= \big(p_{g^0}(X,X') - \tilde{t}_\eta(W,W')\big)\big(\Delta g(X) - \Delta g(X')\big), \tag{68}
$$

which leads to the result

$$
D_g \mathcal{L}^{\mathrm{orth}}(g^0,\eta)[\Delta g] = \mathbb{E}\Big[\big(p_{g^0}(X,X') - \tilde{t}_\eta(W,W')\big)\big(\Delta g(X) - \Delta g(X')\big)\Big]. \tag{69}
$$

**Orthogonality with respect to $\mu_1$.** We now consider perturbations of the nuisance component $\mu_1$ along the path

$$
\mu_{1,s}(x) = \mu_1^0(x) + s\,\Delta\mu_1(x), \qquad \text{with} \quad \Delta\mu_1(x) = \mu_1(x) - \mu_1^0(x), \tag{70}
$$

and write $\eta_s = (\mu_{1,s}, \mu_0^0, e^0)$ such that the directional derivative of interest becomes

$$
D_{\mu_1} D_g \,\mathcal{L}^{\mathrm{orth}}(g^0,\eta^0)[\Delta g, \Delta\mu_1] = \frac{\mathrm{d}}{\mathrm{d}s}\,D_g \mathcal{L}^{\mathrm{orth}}(g^0,\eta_s)[\Delta g]\bigg|_{s=0}, \tag{71}
$$

and, under mild regularity conditions,

$$
D_{\mu_1} D_g \,\mathcal{L}^{\mathrm{orth}}(g^0,\eta^0)[\Delta g, \Delta\mu_1] = \frac{\mathrm{d}}{\mathrm{d}s}\,\mathbb{E}\Big[\big(p_{g^0}(X,X') - \tilde{t}_{\eta_s}(W,W')\big)\big(\Delta g(X) - \Delta g(X')\big)\Big]\bigg|_{s=0} \tag{72}
$$

$$
= -\mathbb{E}\left[\frac{\mathrm{d}}{\mathrm{d}s}\,\tilde{t}_{\eta_s}(W,W')\bigg|_{s=0}\big(\Delta g(X) - \Delta g(X')\big)\right]. \tag{73}
$$

In order to evaluate the inner derivative, we substitute the definition of $\tilde{t}_{\eta_s}(W,W')$ from Equation (60), which yields

$$
\frac{\mathrm{d}}{\mathrm{d}s}\,\tilde{t}_{\eta_s}(W,W')\bigg|_{s=0} = \frac{\mathrm{d}}{\mathrm{d}s}\,t_{\tau_s}(X,X')\bigg|_{s=0}
$$

$$
+ \frac{\mathrm{d}}{\mathrm{d}s}\,\underbrace{\frac{t_{\tau_s}(X,X')\big(1 - t_{\tau_s}(X,X')\big)}{\kappa}}_{A(s)}\,\underbrace{\Big(\big(\phi_{\eta_s}(W) - \tau_s(X)\big) - \big(\phi_{\eta_s}(W') - \tau_s(X')\big)\Big)}_{B(s)}\bigg|_{s=0}. \tag{74}
$$

For the first term, recall that

$$
t_{\tau_s}(X,X') = \sigma\left(\frac{\tau_s(X) - \tau_s(X')}{\kappa}\right), \qquad \text{with} \quad \tau_s(X) = \mu_{1,s}(X) - \mu_0^0(X). \tag{75}
$$

By applying the chain rule, we have

$$
\frac{\mathrm{d}}{\mathrm{d}s}\,t_{\tau_s}(X,X')\bigg|_{s=0} = \frac{t_{\tau^0}(X,X')\big(1 - t_{\tau^0}(X,X')\big)}{\kappa}\big(\Delta\mu_1(X) - \Delta\mu_1(X')\big). \tag{76}
$$

For the second term, we apply the product rule, and for brevity write $\frac{\mathrm{d}}{\mathrm{d}s}A(s)\big|_{s=0} = A'(0)$, such that

$$
\frac{\mathrm{d}}{\mathrm{d}s}\,A(s)B(s)\bigg|_{s=0} = A'(0)B(0) + A(0)B'(0) \tag{77}
$$

$$
= A'(0)\Big(\big(\phi_{\eta^0}(W) - \tau^0(X)\big) - \big(\phi_{\eta^0}(W') - \tau^0(X')\big)\Big)
$$

$$
+ A(0)\frac{\mathrm{d}}{\mathrm{d}s}\Big(\big(\phi_{\eta_s}(W) - \tau_s(X)\big) - \big(\phi_{\eta_s}(W') - \tau_s(X')\big)\Big)\bigg|_{s=0}. \tag{78}
$$

Now, we can substitute the definition of $\phi_{\eta_s}(W)$ from Equation (29) to see that

$$\frac{\mathrm{d}}{\mathrm{d}s}\left(\phi_{\eta_s}(W) - \tau_s(X)\right)\bigg|_{s=0} = \frac{\mathrm{d}}{\mathrm{d}s}\phi_{\eta_s}(W)\bigg|_{s=0} - \frac{\mathrm{d}}{\mathrm{d}s}\tau_s(X)\bigg|_{s=0} \tag{79}$$

$$= \frac{\mathrm{d}}{\mathrm{d}s}\left(\mu_{1,s}(X) - \mu_0^0(X) + \frac{T}{e^0(X)}\left(Y - \mu_{1,s}(X)\right) - \frac{1-T}{1-e^0(X)}\left(Y - \mu_0^0(X)\right)\right)\bigg|_{s=0} - \Delta\mu_1(X) \tag{80}$$

$$= -\frac{T}{e^0(X)}\Delta\mu_1(X). \tag{81}$$

Therefore, the entire second term becomes

$$\frac{\mathrm{d}}{\mathrm{d}s}A(s)B(s)\bigg|_{s=0} = A'(0)\left(\left(\phi_{\eta^0}(W) - \tau^0(X)\right) - \left(\phi_{\eta^0}(W') - \tau^0(X')\right)\right)$$
$$+ \frac{t_{\tau^0}(X,X')\left(1 - t_{\tau^0}(X,X')\right)}{\kappa}\left(\frac{T'}{e^0(X')}\Delta\mu_1(X') - \frac{T}{e^0(X)}\Delta\mu_1(X)\right). \tag{82}$$

Given the expressions for the derivatives of the soft label from Eq. (76) and the correction term from Eq. (82), we obtain

$$\frac{\mathrm{d}}{\mathrm{d}s}\tilde{t}_{\eta_s}(W,W')\bigg|_{s=0} = A'(0)\left(\left(\phi_{\eta^0}(W) - \tau^0(X)\right) - \left(\phi_{\eta^0}(W') - \tau^0(X')\right)\right)$$
$$+ \frac{t_{\tau^0}(X,X')\left(1 - t_{\tau^0}(X,X')\right)}{\kappa}\left(\left(1 - \frac{T}{e^0(X)}\right)\Delta\mu_1(X) - \left(1 - \frac{T'}{e^0(X')}\right)\Delta\mu_1(X')\right). \tag{83}$$

Finally, we can consider the directional derivative of interest in Eq. (73), and apply iterated expectations to yield

$$D_{\mu_1}D_g\mathcal{L}^{\text{orth}}(g^0,\eta^0)[\Delta g, \Delta\mu_1] = -\mathbb{E}_{W,W'}\left[\frac{\mathrm{d}}{\mathrm{d}s}\tilde{t}_{\eta_s}(W,W')\bigg|_{s=0}\left(\Delta g(X) - \Delta g(X')\right)\right] \tag{84}$$

$$= -\mathbb{E}_{X,X'}\left[\mathbb{E}_{T,T',Y,Y'|X,X'}\left[\frac{\mathrm{d}}{\mathrm{d}s}\tilde{t}_{\eta_s}(W,W')\bigg|_{s=0}\right]\left(\Delta g(X) - \Delta g(X')\right)\right] \tag{85}$$

$$= 0, \tag{86}$$

where the last equality follows from

$$\mathbb{E}_{T,Y|X}\left[1 - \frac{T}{e^0(X)}\right] = 0, \qquad \text{and} \quad \mathbb{E}_{T,Y|X}\left[\phi_{\eta^0}(W) - \tau^0(X)\right] = 0, \tag{87}$$

and likewise for the primed counterparts.

This shows that for any perturbation directions $\Delta g$ and $\Delta\mu_1$,

$$D_{\mu_1}D_g\mathcal{L}^{\text{orth}}(g^0,\eta^0)[\Delta g, \Delta\mu_1] = 0. \tag{88}$$

and therefore establishes that the loss $\mathcal{L}^{\text{orth}}$ is Neyman-orthogonal with respect to the nuisance component $\mu_1$.

**Orthogonality with respect to $\mu_0$.** For the nuisance component $\mu_0$, the proof is analogous to the case of $\mu_1$ with some minor differences. We consider perturbations along the path

$$\mu_{0,s}(x) = \mu_0^0(x) + s\,\Delta\mu_0(x), \qquad \text{with} \quad \Delta\mu_0(x) = \mu_0(x) - \mu_0^0(x), \tag{89}$$

and write $\eta_s = (\mu_1^0, \mu_{0,s}, e^0)$. The directional derivative of interest is then

$$D_{\mu_0}D_g\mathcal{L}^{\text{orth}}(g^0,\eta^0)[\Delta g, \Delta\mu_0] = \frac{\mathrm{d}}{\mathrm{d}s}D_g\mathcal{L}^{\text{orth}}(g^0,\eta_s)[\Delta g]\bigg|_{s=0} \tag{90}$$

$$= -\mathbb{E}\left[\frac{\mathrm{d}}{\mathrm{d}s}\tilde{t}_{\eta_s}(W,W')\bigg|_{s=0}\left(\Delta g(X) - \Delta g(X')\right)\right]. \tag{91}$$

By the same steps as in the case of $\mu_1$, but using that $\tau_s(x) = \mu_1^0(x) - \mu_{0,s}(x)$, we get

$$\frac{\mathrm{d}}{\mathrm{d}s}\, \tilde{t}_{\eta_s}(W, W')\bigg|_{s=0} = \frac{t_{\tau^0}(X, X')\big(1 - t_{\tau^0}(X, X')\big)}{\kappa} \left(\Big(\frac{1-T}{1-e^0(X)} - 1\Big)\Delta\mu_0(X) - \Big(\frac{1-T'}{1-e^0(X')} - 1\Big)\Delta\mu_0(X')\right)$$

$$+ \frac{\mathrm{d}}{\mathrm{d}s}\left(\frac{t_{\tau_s}(X, X')\big(1 - t_{\tau_s}(X, X')\big)}{\kappa}\right)\bigg|_{s=0} \Big(\big(\phi_{\eta^0}(W) - \tau^0(X)\big) - \big(\phi_{\eta^0}(W') - \tau^0(X')\big)\Big). \quad (92)$$

We can then express the directional derivative using iterated expectations as

$$D_{\mu_0} D_g \, \mathcal{L}^{\mathrm{orth}}(g^0, \eta^0)[\Delta g, \Delta\mu_0] = -\mathbb{E}_{X,X'}\left[\mathbb{E}_{T,T',Y,Y'|X,X'}\left[\frac{\mathrm{d}}{\mathrm{d}s}\, \tilde{t}_{\eta_s}(W, W')\bigg|_{s=0}\right](\Delta g(X) - \Delta g(X'))\right] \quad (93)$$

$$= 0, \quad (94)$$

since

$$\mathbb{E}_{T,Y|X}\left[\frac{1-T}{1-e^0(X)} - 1\right] = 0, \quad \text{and} \quad \mathbb{E}_{T,Y|X}\Big[\phi_{\eta^0}(W) - \tau^0(X)\Big] = 0, \quad (95)$$

and likewise for the primed counterparts. This derivation establishes orthogonality of $\mathcal{L}^{\mathrm{orth}}$ with respect to the nuisance component $\mu_0$.

### Orthogonality with respect to $e$.

For the propensity score $e$, the orthogonality argument follows the same overall strategy as for $\mu_1$ and $\mu_0$, but the dependence of $\tilde{t}_\eta$ on $e$ enters only through the doubly robust score $\phi_\eta$ (Eq. (29)). We consider perturbations of $e$ along the path

$$e_s(x) = e^0(x) + s\,\Delta e(x), \quad \text{with} \quad \Delta e(x) = e(x) - e^0(x), \quad (96)$$

and define $\eta_s = (\mu_1^0, \mu_0^0, e_s)$. The directional derivative of interest is then

$$D_e D_g \, \mathcal{L}^{\mathrm{orth}}(g^0, \eta^0)[\Delta g, \Delta e] = \frac{\mathrm{d}}{\mathrm{d}s}\, D_g \mathcal{L}^{\mathrm{orth}}(g^0, \eta_s)[\Delta g]\bigg|_{s=0} \quad (97)$$

$$= -\mathbb{E}\left[\frac{\mathrm{d}}{\mathrm{d}s}\, \tilde{t}_{\eta_s}(W, W')\bigg|_{s=0} (\Delta g(X) - \Delta g(X'))\right]. \quad (98)$$

Now, by relying on the definition of $\tilde{t}_{\eta_s}(W, W')$ from Equation (60), the inner derivative simplifies to

$$\frac{\mathrm{d}}{\mathrm{d}s}\, \tilde{t}_{\eta_s}(W, W')\bigg|_{s=0} = \frac{t_{\tau^0}(X, X')\big(1 - t_{\tau^0}(X, X')\big)}{\kappa} \cdot \frac{\mathrm{d}}{\mathrm{d}s}\Big(\big(\phi_{\eta_s}(W) - \tau^0(X)\big) - \big(\phi_{\eta_s}(W') - \tau^0(X')\big)\Big)\bigg|_{s=0} \quad (99)$$

$$= \frac{t_{\tau^0}(X, X')\big(1 - t_{\tau^0}(X, X')\big)}{\kappa} \left(\frac{\mathrm{d}}{\mathrm{d}s}\phi_{\eta_s}(W)\bigg|_{s=0} - \frac{\mathrm{d}}{\mathrm{d}s}\phi_{\eta_s}(W')\bigg|_{s=0}\right), \quad (100)$$

since only the doubly robust score $\phi_{\eta_s}$ depends on $s$.

If we now focus on the derivative of this doubly robust score, we get

$$\frac{\mathrm{d}}{\mathrm{d}s}\phi_{\eta_s}(W)\bigg|_{s=0} = \frac{\mathrm{d}}{\mathrm{d}s}\left(\mu_1^0(X) - \mu_0^0(X) + \frac{T}{e_s(X)}\big(Y - \mu_1^0(X)\big) - \frac{1-T}{1-e_s(X)}\big(Y - \mu_0^0(X)\big)\right)\bigg|_{s=0} \quad (101)$$

$$= -T\big(Y - \mu_1^0(X)\big)\frac{\Delta e(X)}{e^0(X)^2} - (1-T)\big(Y - \mu_0^0(X)\big)\frac{\Delta e(X)}{\big(1 - e^0(X)\big)^2} \quad (102)$$

$$= -\Delta e(X)\left(\frac{T\big(Y - \mu_1^0(X)\big)}{e^0(X)^2} + \frac{(1-T)\big(Y - \mu_0^0(X)\big)}{\big(1 - e^0(X)\big)^2}\right). \quad (103)$$

We can again express the directional derivative using iterated expectations as

$$D_e D_g \mathcal{L}^{\text{orth}}(g^0, \eta^0)[\Delta g, \Delta e] = -\mathbb{E}_{X,X'}\left[\mathbb{E}_{T,T',Y,Y'|X,X'}\left[\frac{\mathrm{d}}{\mathrm{d}s}\tilde{t}_{\eta_s}(W, W')\Big|_{s=0}\right](\Delta g(X) - \Delta g(X'))\right] \tag{104}$$

$$= 0, \tag{105}$$

where the last equality follows from

$$\mathbb{E}_{T,T',Y,Y'|X,X'}\left[\frac{T(Y - \mu_1^0(X))}{e^0(X)^2} + \frac{(1-T)(Y - \mu_0^0(X))}{\left(1 - e^0(X)\right)^2}\right] \tag{106}$$

$$= \frac{1}{e^0(X)^2}\ \mathbb{E}_{T,Y|X}\left[T(Y - \mu_1^0(X))\right] + \frac{1}{\left(1 - e^0(X)\right)^2}\ \mathbb{E}_{T,Y|X}\left[(1-T)(Y - \mu_0^0(X))\right] \tag{107}$$

$$= 0, \tag{108}$$

and likewise for the corresponding primed term, which therefore establishes that $\mathcal{L}^{\text{orth}}$ is Neyman-orthogonal with respect to $e$.

**Conclusion.** We have shown that for each nuisance component $\eta = (\mu_1, \mu_0, e)$ and for arbitrary perturbation directions $(\Delta g, \Delta \eta)$, the cross-derivative of the loss satisfies

$$D_{\mu_1} D_g \mathcal{L}^{\text{orth}}(g^0, \eta^0)[\Delta g, \Delta \mu_1] = D_{\mu_0} D_g \mathcal{L}^{\text{orth}}(g^0, \eta^0)[\Delta g, \Delta \mu_0] = D_e D_g \mathcal{L}^{\text{orth}}(g^0, \eta^0)[\Delta g, \Delta e] = 0. \tag{109}$$

Therefore, the loss $\mathcal{L}^{\text{orth}}$ is Neyman-orthogonal with respect to the nuisance components $(\mu_1, \mu_0, e)$. $\qquad\square$

# D. Population optima of the considered learning objectives

In this section, we characterize the population optima of our considered losses at the true nuisances $\eta^0$. We show that all losses aim to recover a scoring function $g(x)$ whose ranking matches that of the conditional average treatment effect $\tau(x)$, but they differ in how strongly they constrain $g$. Therefore, every population-optimal solution must preserve the $\tau$-induced ordering, and the losses only differ in how much additional structure they impose on $g(x)$. We then discuss the role of the smoothing parameter $\kappa$ in the orthogonal ranking loss $\mathcal{L}^{\text{orth}}$.

### D.1. Population optima

**Minimizer of the mean squared error loss.** The simplest objective that we consider is the mean squared error between the scoring function $g(x)$ and the true CATE $\tau^0(x)$. It is given by

$$\mathcal{L}^{\text{cate}}(g, \eta^0) = \mathbb{E}_X \left[ \left( g(X) - \tau^0(X) \right)^2 \right],\tag{110}$$

and is uniquely minimized at $g^0(x) = \tau^0(x)$. Therefore, the minimizer preserves the $\tau$-induced ordering as it fully identifies the CATE. Minimizing this loss requires recovering the full structure of $\tau(x)$ and not just the ranking.

**Population optima of the binary ranking loss.** We next consider the binary ranking loss, which is binary cross-entropy between the pairwise probability predictions of the model $p_g(x, x')$ and the binary indicators of the $\tau$-induced ordering. It is given by

$$\mathcal{L}^{\text{bin}}(g, \eta^0) = \mathbb{E}_{X,X'} \left[ - b_{\tau^0}(X, X') \log p_g(X, X') - \left( 1 - b_{\tau^0}(X, X') \right) \log \left( 1 - p_g(X, X') \right) \right],\tag{111}$$

where

$$p_g(X, X') = \sigma\left( g(X) - g(X') \right), \qquad \text{and} \quad b_{\tau^0}(X, X') = \mathbf{I} \left\{ \tau^0(X) > \tau^0(X') \right\}.\tag{112}$$

If we consider a given pair $(x, x')$, the infimum of the loss contribution equals zero and is approached when the model probability $p_g(x, x')$ converges to the binary label $b_{\tau^0}(x, x')$. This occurs as the margin $g(x) - g(x')$ tends to $+\infty$ if $b_{\tau^0}(x, x') = 1$ and to $-\infty$ otherwise. For any finite model $g$, the infimum cannot be attained, but correctly ordered pairs can have a loss contribution arbitrarily close to zero. In contrast, if a candidate model $g$ misorders a pair, the margin $g(x) - g(x')$ will have the wrong sign, and the loss contribution remains strictly bounded away from zero with a positive constant. Therefore, if $g$ does not follow the $\tau$-induced ordering, there must be at least one such misordered pair keeping the population loss bounded away from its infimum. If $g$ does respect the $\tau$-induced ordering, all pairwise loss contributions can be made arbitrarily small by scaling $g$. It follows that any population-optimal solution of $\mathcal{L}^{\text{bin}}$, in the sense of achieving a risk arbitrarily close to the infimum, must preserve the ordering of the treatment effects.

The binary ranking loss does not admit a unique population minimizer, as its infimum cannot be attained by a finite model $g$. Once the model $g$ agrees with the ordering induced by $\tau$, scaling $g$ by any positive constant increases all pairwise margins and drives the loss arbitrarily close to zero. Additionally, as the loss depends only on the difference $g(x) - g(x')$, adding a constant shift to $g$ leaves all pairwise predictions $p_g(x, x')$, and thus the total loss, unchanged. As a result, any scoring function $g$ that follows the $\tau$-ordering can be considered population-optimal. These are all functions of the form $g^0(x) = h\left( \tau^0(x) \right)$ for some strictly increasing transformation $h$. This invariance is precisely why learning a ranking of treatment effects is an easier problem than learning the exact treatment effect magnitudes.

**Minimizers of the soft ranking loss.** We now consider the soft ranking loss, which is binary cross-entropy between the pairwise probability predictions of the model $p_g(x, x')$ and smoothed pairwise targets based on the CATE differences. The loss is given by

$$\mathcal{L}^{\text{soft}}(g, \eta^0) = \mathbb{E}_{X,X' \sim \mathbb{P}(X)} \left[ - t_{\tau^0}(X, X') \log p_g(X, X') - \left( 1 - t_{\tau^0}(X, X') \right) \log \left( 1 - p_g(X, X') \right) \right],\tag{113}$$

where

$$p_g(X, X') = \sigma\left( g(X) - g(X') \right), \qquad \text{and} \quad t_{\tau^0}(X, X') = \sigma \left( \frac{\tau^0(X) - \tau^0(X')}{\kappa} \right),\tag{114}$$

with $\kappa > 0$ a smoothing parameter.

Unlike the binary ranking targets, the soft targets $t_{\tau^0}(x, x')$ do not only encode the ordering of $\tau^0(x) - \tau^0(x')$, but also the magnitude of this difference, scaled by $(1/\kappa)$. The loss contribution for a pair $(x, x')$ is exactly minimized when the model probability $p_g(x, x')$ matches the soft targets $t_{\tau^0}(x, x')$, which occurs precisely when

$$g(x) - g(x') = \frac{\tau^0(x) - \tau^0(x')}{\kappa}. \tag{115}$$

This constraint is satisfied jointly for all pairs when the scoring model takes the form

$$g^0(x) = \frac{1}{\kappa}\tau^0(x) + c, \tag{116}$$

for some constant $c \in \mathbb{R}$. In contrast to the binary ranking loss, the soft ranking loss identifies $\tau(x)$ up to an additive constant and a fixed scaling factor $(1/\kappa)$, imposing more structure on $g$ than just the ranking.

**Minimizers of the orthogonal loss.** We finally consider the orthogonal ranking loss, which was derived based on the influence function of the soft ranking loss. The population objective is given by

$$\mathcal{L}^{\text{orth}}(g, \eta^0) = \mathbb{E}_{W,W'}\Big[ - \tilde{t}_{\eta^0}(W, W') \log p_g(X, X') - \big(1 - \tilde{t}_{\eta^0}(W, W')\big) \log\big(1 - p_g(X, X')\big)\Big], \tag{117}$$

where $p_g(X, X') = \sigma\big(g(X) - g(X')\big)$, and where the pseudo labels are defined as

$$\tilde{t}_{\eta^0}(W, W') = t_{\tau^0}(X, X') + \frac{t_{\tau^0}(X, X')\big(1 - t_{\tau^0}(X, X')\big)}{\kappa}\Big(\big(\phi_{\eta^0}(W) - \tau^0(X)\big) - \big(\phi_{\eta^0}(W') - \tau^0(X')\big)\Big). \tag{118}$$

Since the orthogonal loss is also a binary cross-entropy objective, the loss contribution of a pair $(x, x')$ is minimized when the model probability $p_g(x, x')$ matches the conditional expectation of the pseudo labels given $X = x$ and $X = x'$, or

$$p_g(x, x') = \mathbb{E}\big[\,\tilde{t}_{\eta^0}(W, W') \mid X = x, X' = x'\big]. \tag{119}$$

To evaluate this expectation, we can rely on the property of the doubly robust score, i.e.,

$$\mathbb{E}\big[\phi_{\eta^0}(W) - \tau^0(X) \mid X\big] = 0, \tag{120}$$

and likewise for the primed counterpart, which implies that the correction term in $\tilde{t}_{\eta^0}(W, W')$ has zero conditional mean. Consequently, the loss contribution is minimized when

$$p_g(x, x') = \mathbb{E}\big[\,\tilde{t}_{\eta^0}(W, W') \mid X = x, X' = x'\big] = t_{\tau^0}(x, x'). \tag{121}$$

This means that the correction based on the influence function does not change the population minimizers, and the orthogonal loss has the same minimizers as the soft ranking loss

$$g^0(x) = \frac{1}{\kappa}\tau^0(x) + c. \tag{122}$$

### D.2. Interpretation of the smoothness parameter $\kappa$

Now that we have derived the population optima of the considered loss functions, we discuss the behavior of the orthogonal ranking loss as a function of the smoothness parameter $\kappa$. While, for any fixed and finite $\kappa > 0$, the population minimizers preserve the ordering of the treatment effects, the choice of $\kappa$ strongly influences the shape of the loss landscape and the structure imposed on $g$.

**Small $\kappa$.** As $\kappa \to 0$, the soft targets converge to the targets of the binary ranking loss, which yields

$$t_{\tau^0}(X, X') = \sigma\left(\frac{\tau^0(X) - \tau^0(X')}{\kappa}\right) \quad \longrightarrow \quad b_{\tau^0}(X, X') = \mathbf{I}\big\{\tau^0(X) > \tau^0(X')\big\}. \tag{123}$$

Accordingly, both the soft and orthogonal ranking losses ($\mathcal{L}^{\text{soft}}$ and $\mathcal{L}^{\text{orth}}$) approach the binary ranking loss ($\mathcal{L}^{\text{bin}}$). At the same time, their population minimizers $g^0(x) = \tau^0(x)/\kappa + c$ become unbounded, reflecting that the binary ranking loss

does not admit a finite population minimizer. In this regime, any scoring function that preserves the ordering induced by $\tau^0(x)$ can achieve a population risk arbitrarily close to the infimum.

This behavior can also be understood through the geometry of these losses. At the population optimum, the second derivative of a single pairwise loss contribution (of both $\mathcal{L}^{\text{soft}}$ and $\mathcal{L}^{\text{orth}}$), with respect to the margin $g(x) - g(x')$, is given by

$$\nabla^2_{g(x)-g(x')} \ell(g^0, \eta^0) = p_{g^0}(x, x')\big(1 - p_{g^0}(x, x')\big) \tag{124}$$

$$= \sigma\left(\frac{\tau^0(x) - \tau^0(x')}{\kappa}\right)\left(1 - \sigma\left(\frac{\tau^0(x) - \tau^0(x')}{\kappa}\right)\right). \tag{125}$$

As the smoothing parameter $\kappa$ goes to zero, the loss becomes locally flat along directions that preserve the $\tau$-induced ordering. In the limit, the learning problem therefore reduces to recovering the ranking of the treatment effects, rather than the exact magnitudes of their differences.

**Intermediate $\kappa$.** For $\kappa = 1$, the soft targets become $t_{\tau^0}(X, X') = \sigma\left(\tau^0(X) - \tau^0(X')\right)$, and encode both the ordering and the relative magnitude of the treatment effect differences. The population minimizer of the soft and orthogonal ranking losses then takes the form $g^0(x) = \tau^0(x) + c$, implying that the learning problem requires recovering the full functional form of $\tau^0(x)$, rather than only the ordering. This behavior is also reflected in the curvature of the loss around the optimum (Eq. (125)) which is now strictly positive at the population minimizer and penalizes deviations of the margins $g(x) - g(x')$ from $\tau^0(x) - \tau^0(x')$. This choice of parameter $\kappa$ therefore enforces the most structure on the scoring function $g$.

**Large $\kappa$.** As $\kappa \to \infty$, the soft targets satisfy

$$t_{\tau^0}(X, X') = \sigma\left(\frac{\tau^0(X) - \tau^0(X')}{\kappa}\right) \longrightarrow \frac{1}{2}, \tag{126}$$

and therefore become independent of the ordering and the magnitude of the treatment effect differences. Consequently, the population minimizer of the soft and orthogonal losses becomes $g^0(x) = c$, with the model predictions collapsing to $p_{g^0}(x, x') = 1/2$. In this regime, where $\kappa \to \infty$, all ordering information is lost, which enforces a constant scoring function $g(x)$. This arises only in the limit, and for any fixed and finite $\kappa > 0$, both the soft and orthogonal targets remain dependent on $\tau^0(x) - \tau^0(x')$ and therefore preserve the ordering of treatment effects.

**Choice of $\kappa$.** Since our objective is purely ranking of treatment effects, the discussion above suggests choosing $\kappa$ to be small. At the population level, this reduces the learning problem to recovering only the $\tau$-induced ordering. In practice however, the pairwise targets $t_{\tau^0}(x, x')$ or $b_{\tau^0}(x, x')$ are not observed and must be estimated from data. Such estimates can introduce plug-in bias, which is precisely the issue addressed by the orthogonal loss $\mathcal{L}^{\text{orth}}$.

At the same time, as $\kappa$ decreases, the orthogonal correction amplifies for pairs with similar treatment effects, which increases the variability of $\tilde{t}_\eta(w, w')$ in finite samples. Consequently, while smaller values of $\kappa$ impose weaker structural constraints on $g$ and move the problem closer to pure ranking, they also amplify the estimation noise in finite samples. The choice of $\kappa$ thus reflects a bias-variance trade-off. We propose to select $\kappa$ using an approximation of the AUTOC criterion on a validation set, which aligns the model selection directly with the ranking objective (Chernozhukov et al., 2025).

# E. Experimental setup

This appendix provides additional information on the experimental setup and data-generating processes used in our experiments. We consider both a fully synthetic benchmark, where covariates are generated from known distributions, and semi-synthetic benchmarks, where real-world covariates are combined with simulated treatments and outcomes. We additionally benchmark *Rank-Learner* on the real-world CRITEO uplift dataset. Together, these experiments allow us to evaluate *Rank-Learner* under controlled conditions as well as realistic feature distributions and observational settings. The full implementation of all data-generating processes and experiments is available at https://github.com/henriarnoUG/rank-learner.

## E.1. Synthetic data-generating process

We begin our experiments on a fully synthetic dataset, which enables a controlled evaluation of our proposed *Rank-Learner*. The data-generating process is adapted from prior work (Kamran et al., 2024) and is constructed such that the treatment effects are generated as a strictly increasing, non-linear transformation of a latent score $s(x)$. As a result, the ranking of treatment effects is fully determined by $s(x)$, while their magnitudes depend on the transformation, allowing us to decouple the ranking problem from pointwise effect estimation. The treatment assignment depends on $s(x)$ and an independent component (to induce confounding while maintaining overlap), and the baseline outcome $\mu_0(x)$ is generated partly from covariates not used in $s(x)$, such that outcome prediction does not fully determine the treatment effect ranking.

The covariates $X \in \mathbb{R}^{10}$ are sampled i.i.d. from a multivariate standard normal distribution,

$$X \sim \mathcal{N}(0, I_{10}). \tag{127}$$

We define a latent score $s(X)$ that fully determines the ranking of treatment effects as

$$s(X) = 0.8X_1 + 0.6X_2 + 0.4X_3 + 0.3X_1^2 - 0.2X_2X_3. \tag{128}$$

The treatment effects are generated as strictly increasing, non-linear transformations of $s(X)$, i.e.,

$$\tau(X) = s(X) + 0.5\tanh\big(s(X)\big). \tag{129}$$

The nuisance components (propensity score and potential outcomes) are generated as

$$e(X) = \sigma\Big(0.8\,s(X) + 0.6\big(X_6 - 0.5X_7\big)\Big), \tag{130}$$

$$\mu_0(X) = 0.5X_2 - 0.4X_3 + 0.3\sin(X_4) + 0.2\big(X_5^2 - 1\big), \tag{131}$$

$$\mu_1(X) = \mu_0(X) + \tau(X) \tag{132}$$

where $\sigma(.)$ denotes the logistic sigmoid.

The treatment indicators and observed outcomes are sampled as

$$T \sim \text{Bernoulli}\big(e(X)\big) \tag{133}$$

$$Y = T\,\mu_1(X) + (1 - T)\,\mu_0(X) + \varepsilon, \qquad \varepsilon \sim \mathcal{N}(0, 0.6^2). \tag{134}$$

For all synthetic experiments, we generate a fixed test set of 1,000 samples that is exclusively used for evaluation. Training data are generated with varying sample sizes (ranging from 100 to 2,000 samples). This allows us to study the ranking performance as a function of training size, which directly affects the quality of the estimated nuisance components. For each considered training size $n$, we use sample splitting: $n$ samples are used to estimate the nuisance components (of which 80% are used to fit the models and 20% for validation), and an independent set of $n$ samples is used to train the second-stage models (again split into 80% for training and 20% for validation). All experiments are repeated over five random seeds, which affect both data sampling and weight initialization of the models.

### E.2. Semi-synthetic data-generating processes

In the semi-synthetic setting, we use real-world covariates from three established datasets covering distinct application domains: MOVIELENS (recommender systems) (Harper & Konstan, 2015), MIMIC-III (healthcare) (Johnson et al., 2016), and the CURRENT POPULATION SURVEY (public policy) (Flood et al., 2025). This preserves realistic feature distributions while still enabling controlled evaluation with access to ground-truth treatment effects. The exact data-generating processes can be found in the accompanying code repository.

Across the three semi-synthetic datasets, we follow the same data-generating structure as in the fully synthetic setting. In particular, treatment effects are generated as strictly increasing, non-linear transformations of a latent score $s(x)$, such that the ranking of treatment effects is fully determined by $s(x)$, while the magnitudes depend on the transformation. Treatment assignment depends on $s(x)$ and an independent component to induce confounding while maintaining overlap, and $\mu_0(x)$ is generated only partly from $s(x)$. All components of the generated data are designed to reflect realistic, domain-specific mechanisms for the considered applications.

For all semi-synthetic experiments, we generate a fixed test set of 1,000 samples that is used exclusively for evaluation. As in the synthetic setting, we perform sample splitting with nuisance components and second-stage models trained on separate samples. We consider a fixed training size of 1,000 samples, of which 80% is used for training and 20% for validation. All experiments are repeated over five random seeds, which affect both data sampling and weight initialization.

• **MOVIELENS** (Harper & Konstan, 2015). From this dataset, we extract user-level covariates capturing demographics (age, gender, and occupation), behavioral traits (mean and standard deviation of past movie ratings), and features capturing movie preference (genre-specific share of watched movies). We simulate a setting where the treatment corresponds to exposure to a targeted advertisement on a movie platform, and the outcome represents user engagement. Treatment assignment is biased towards specific occupational groups, in particular students, and the treatment effects are primarily driven by user preferences, reflecting that users whose preferred genres align more closely with the advertised content respond more positively. Baseline engagement is mostly driven by the historical rating behavior and is weakly correlated with the treatment effects.

• **MIMIC-III** (Johnson et al., 2016). From this dataset, we extract patient-level covariates capturing demographics (age and gender) and vital sign measurements (heart rate, blood pressure, arterial pressure, respiratory rate, and oxygen saturation). We simulate a clinical decision-making setting in which a medical treatment is administered that affects patient recovery (the outcome reflects a continuous recovery score). The treatment effects are primarily driven by instability in blood circulation (based on abnormal blood pressure and arterial pressure measurements), reflecting that patients with a more severe condition can benefit most from treatment. Treatment assignment is biased towards patients exhibiting signs of acute physiological distress, such as an elevated heart rate and respiratory rate (while also depending on blood circulation to induce confounding). Baseline recovery is primarily driven by fever and oxygen saturation and is only mildly correlated with the treatment effects.

• **CPS** (Flood et al., 2025). From this dataset, we extract individual-level covariates capturing demographics (age, gender, and educational attainment), household structure (marital status and number of dependents), and labor-market characteristics (hours worked per week, weeks worked per year, firm size, wage, and household income). We simulate a policy intervention (representing, for instance, a job-training program), with the outcome representing a continuous measure of future earnings potential. Treatment effects are primarily driven by individual skill readiness and economic need (higher effects for younger individuals with a low current income and a high educational attainment). Treatment assignment follows a household-based eligibility heuristic that prioritizes individuals with dependents and greater family responsibilities. Baseline outcomes are driven mainly by employment intensity and current income and are moderately correlated with the treatment effects.

### E.3. Real-world uplift benchmark

We complement the synthetic and semi-synthetic experiments with the real-world CRITEO uplift benchmark (Diemert et al., 2021), which contains randomized data from online advertising campaigns. Each observation consists of user covariates $X \in \mathbb{R}^{12}$, a treatment indicator $T \in \{0, 1\}$ corresponding to ad exposure, and a binary outcome $Y \in \{0, 1\}$ indicating whether the user visited the advertiser's website. The treatment effects therefore capture how ad exposure changes the probability of visiting the website.

To evaluate *Rank-Learner* on CRITEO in an observational setting, we follow Diemert et al. (2021) and induce confounding in the training data through selective subsampling, while leaving the test data randomized. Starting from a large randomized training pool, we define a covariate-dependent sampling probability

$$p(x) = \left(1 - 2\delta\right) \sigma \left( \sum_j x_j \right) + \delta, \qquad \delta = 0.05, \tag{135}$$

where the summation is taken over the five most predictive covariates for the outcome and $\sigma(\cdot)$ denotes the logistic sigmoid. We then retain treated observations with probability $p(x)$ and control observations with probability $1 - p(x)$. This induces dependence between treatment assignment and covariates in the training data while preserving the original observed $(X, T, Y)$ triplets. In practice, a classifier predicting treatment assignment from the covariates achieves an AUC of 0.67 on the confounded training data, compared to 0.50 on the untouched randomized test data.

Since ground-truth treatment effects are unknown in the real-world setting, we evaluate ranking quality using the *area under the uplift curve* (AUUC), computed on untouched randomized test data. AUUC measures how well a model ranks individuals according to their treatment effect: a strong uplift model assigns high scores to individuals who benefit most from treatment, such that the treated-control outcome difference is largest among the highest-ranked users. Following Diemert et al. (2021), we use the *separate relative AUUC* metric, which is specifically designed to remain robust under the substantial treatment imbalance present in the CRITEO benchmark. Due to the rare outcomes and relatively weak uplift signal in this benchmark, we report results on randomized test sets of varying size (50k, 500k, and 1M observations) to reduce evaluation noise. Unlike the synthetic and semi-synthetic experiments, we sample a new randomized test set for each seed rather than evaluating on a fixed test split. We estimate the nuisance functions using the same network architectures and training procedure as in the synthetic and semi-synthetic experiments. Since the outcome is binary and highly imbalanced, the response models are trained using weighted binary cross-entropy losses. We use sample splitting with fixed train and validation sizes of 10,000 and 5,000 samples, respectively.

### E.4. Implementation details

This section provides implementation details shared across all experiments. The same architectures and training procedures are used across datasets and model components. All experiments are conducted on a single machine equipped with an NVIDIA GeForce GTX 2080 GPU, and training is computationally lightweight, with most models converging within a few minutes. All nuisance functions, CATE estimators, and rankers are implemented as feedforward neural networks with a single hidden layer and ReLU activations. Output layers are linear for regression tasks and sigmoid for binary classification tasks. Models are trained using the Adam optimizer with a maximum of 50 epochs and early stopping based on the validation loss (patience of five epochs), retaining the parameters corresponding to the best validation performance.

Hyperparameters are minimally tuned once using a small random search (12 runs) on a validation dataset and then fixed across all experiments and seeds. The search space includes hidden layer width $\in \{64, 128\}$, learning rate $\in \{10^{-4}, 3 \cdot 10^{-4}, 5 \cdot 10^{-4}, 10^{-3}\}$, weight decay $\in \{0, 10^{-5}, 10^{-4}\}$, and batch size $\in \{128, 256\}$. For *Rank-Learner* and the plug-in ranker, the smoothness parameter $\kappa \in \{0.25, 0.5, 1, 1.5, 3\}$ is additionally tuned using the approximated validation AUTOC. Finally, since the orthogonal correction in the *Rank-Learner* pseudo labels can occasionally yield values outside the $(0, 1)$ interval, we clip these labels to $(0, 1)$ as a simple stabilization strategy. We study the impact of this clipping procedure in Appendix F.3.

# F. Extended experimental results

This section reports extended experimental results complementing the main paper. We first present the complete results for the synthetic, semi-synthetic, and real-world benchmarks. We then provide additional analyses on computational efficiency, robustness and sensitivity, and comparisons to additional baselines and benchmarks.

## F.1. Main experimental results

This subsection reports the complete experimental results complementing the findings presented in the main paper. In addition to AUTOC and AUUC, we report the mean policy value $\overline{V}$, which measures the average outcome under the implied policy that treats the top-ranked individuals according to the learned ranking function $\hat{g}$.

### F.1.1. Synthetic benchmark results

Table 6 reports AUTOC and mean policy value across training sizes on the synthetic benchmark. Across both metrics, *Rank-Learner* consistently achieves the strongest performance among the considered baselines. The benefits over the non-orthogonal plug-in ranker are largest in small-sample regimes, where nuisance estimation is most difficult, supporting the theoretical motivation for orthogonalization. As the training size increases, performance differences between methods gradually decrease and all approaches converge toward the oracle performance, which aligns with the improving nuisance estimation quality reported in Table 7. Overall, these results demonstrate that directly optimizing an orthogonal ranking objective leads to robust ranking performance across sample sizes.

*Table 6.* **Synthetic benchmark (main results).** Test AUTOC and mean policy value (mean $\pm$ std dev over five seeds) across training sizes. *Higher is better*. The oracle column reports the metrics obtained by ranking the test set using the true treatment effects. Best mean is shown in **bold**.

| Metric | Method | $n = 100$ | $n = 250$ | $n = 500$ | $n = 1,000$ | $n = 2,000$ | oracle |
|---|---|---|---|---|---|---|---|
| AUTOC ($\uparrow$) | T-learner | $0.88 \pm 0.17$ | $0.96 \pm 0.14$ | $1.24 \pm 0.05$ | $1.32 \pm 0.02$ | $1.36 \pm 0.00$ | |
| | DR-learner | $0.80 \pm 0.18$ | $1.16 \pm 0.12$ | $1.28 \pm 0.05$ | $1.33 \pm 0.02$ | $1.36 \pm 0.02$ | 1.40 |
| | Plug-in ranker | $0.69 \pm 0.32$ | $0.95 \pm 0.14$ | $1.24 \pm 0.06$ | $1.31 \pm 0.02$ | $1.36 \pm 0.00$ | |
| | Rank-learner (*ours*) | $\mathbf{1.00 \pm 0.19}$ | $\mathbf{1.28 \pm 0.03}$ | $\mathbf{1.31 \pm 0.01}$ | $\mathbf{1.34 \pm 0.01}$ | $\mathbf{1.37 \pm 0.00}$ | |
| $\overline{V}$ ($\uparrow$) | T-learner | $0.46 \pm 0.04$ | $0.49 \pm 0.03$ | $0.56 \pm 0.01$ | $0.57 \pm 0.01$ | $0.58 \pm 0.00$ | |
| | DR-learner | $0.42 \pm 0.05$ | $0.53 \pm 0.03$ | $0.55 \pm 0.02$ | $0.57 \pm 0.00$ | $0.58 \pm 0.00$ | 0.59 |
| | Plug-in ranker | $0.38 \pm 0.09$ | $0.48 \pm 0.03$ | $0.56 \pm 0.01$ | $0.57 \pm 0.01$ | $0.58 \pm 0.00$ | |
| | Rank-learner (*ours*) | $\mathbf{0.47 \pm 0.06}$ | $\mathbf{0.55 \pm 0.01}$ | $\mathbf{0.56 \pm 0.01}$ | $\mathbf{0.57 \pm 0.00}$ | $\mathbf{0.58 \pm 0.00}$ | |

*Table 7.* **Synthetic benchmark (nuisance estimation).** Nuisance estimation quality across training sizes, evaluated on the test set (mean $\pm$ std dev over seeds). MSE denotes mean squared error and BCE denotes binary cross-entropy. *Lower is better.*

| Metric | $n = 100$ | $n = 250$ | $n = 500$ | $n = 1,000$ | $n = 2,000$ |
|---|---|---|---|---|---|
| MSE ($\mu_0$) | $0.30 \pm 0.09$ | $0.24 \pm 0.06$ | $0.14 \pm 0.03$ | $0.11 \pm 0.04$ | $0.08 \pm 0.03$ |
| MSE ($\mu_1$) | $1.72 \pm 0.28$ | $1.66 \pm 0.23$ | $0.78 \pm 0.22$ | $0.35 \pm 0.05$ | $0.18 \pm 0.03$ |
| BCE ($e$) | $0.689 \pm 0.004$ | $0.685 \pm 0.009$ | $0.678 \pm 0.011$ | $0.667 \pm 0.003$ | $0.663 \pm 0.002$ |

### F.1.2. Semi-synthetic benchmark results

Table 8 reports AUTOC and mean policy value on the semi-synthetic benchmarks, where real-world covariates are combined with simulated treatments and outcomes. Across all datasets and metrics, *Rank-Learner* achieves the strongest performance among the considered baselines. The improvements over the non-orthogonal plug-in ranker remain consistent across datasets, indicating that the advantages of orthogonal ranking extend beyond fully synthetic settings to realistic feature distributions.

### F.1.3. Real-world uplift benchmark results

Table 9 reports AUUC on the CRITEO uplift benchmark across randomized test sets of increasing size (50k, 500k, 1M). Across all test sizes, *Rank-Learner* achieves the strongest mean AUUC among the considered baselines. Overall, these results support the benefit of directly optimizing an orthogonal ranking objective under confounded training in realistic uplift modeling settings.

*Table 8.* **Semi-synthetic benchmarks.** Test AUTOC and mean policy value (mean $\pm$ std dev over five seeds) with training size $n = 1,000$. *Higher is better.* The oracle column reports the metrics obtained by ranking the test set using the true treatment effects. Best mean is shown in **bold**.

| Metric | Method | MOVIELENS | MIMIC-III | CPS |
|---|---|---|---|---|
| AUTOC ($\uparrow$) | T-learner | $1.31 \pm 0.03$ | $1.12 \pm 0.05$ | $0.87 \pm 0.08$ |
| | DR-learner | $1.34 \pm 0.02$ | $1.17 \pm 0.02$ | $0.92 \pm 0.02$ |
| | Plug-in ranker | $1.30 \pm 0.03$ | $1.11 \pm 0.05$ | $0.87 \pm 0.08$ |
| | Rank-learner (*ours*) | $\mathbf{1.35 \pm 0.01}$ | $\mathbf{1.18 \pm 0.02}$ | $\mathbf{0.95 \pm 0.01}$ |
| | oracle | $1.39$ | $1.22$ | $1.01$ |
| $\overline{V}$ ($\uparrow$) | T-learner | $0.78 \pm 0.01$ | $0.95 \pm 0.02$ | $1.43 \pm 0.03$ |
| | DR-learner | $0.79 \pm 0.00$ | $0.97 \pm 0.00$ | $1.45 \pm 0.01$ |
| | Plug-in ranker | $0.78 \pm 0.01$ | $0.95 \pm 0.02$ | $1.43 \pm 0.03$ |
| | Rank-learner (*ours*) | $\mathbf{0.79 \pm 0.00}$ | $\mathbf{0.97 \pm 0.00}$ | $\mathbf{1.46 \pm 0.00}$ |
| | oracle | $0.80$ | $0.98$ | $1.48$ |

*Table 9.* **Criteo uplift benchmark.** AUUC on randomized test sets of increasing size (50k, 500k, and 1M). Results are reported as mean $\pm$ std dev over five seeds. Values are reported $\times 10^3$ for readability. *Higher is better.* Best mean is shown in **bold**.

| Method | AUUC $\times 10^3$ | | |
|---|---|---|---|
| | 50k | 500k | 1M |
| T-learner | $3.74 \pm 1.18$ | $5.09 \pm 1.59$ | $5.08 \pm 1.62$ |
| DR-learner | $4.44 \pm 1.12$ | $5.01 \pm 1.04$ | $5.17 \pm 1.13$ |
| Plug-in ranker | $3.78 \pm 1.59$ | $4.99 \pm 1.69$ | $5.04 \pm 1.65$ |
| Rank-learner (*ours*) | $\mathbf{5.19 \pm 1.87}$ | $\mathbf{5.83 \pm 0.57}$ | $\mathbf{5.90 \pm 0.40}$ |

## F.2. Computational efficiency

### F.2.1. SUBSAMPLING TRAINING PAIRS

Figure 5 studies the effect of subsampling training pairs on the ranking performance of *Rank-Learner* on the synthetic benchmark. In this experiment, we fix the training size to $n = 1,000$ and estimate the nuisance components using sample splitting. The second-stage ranker is then trained using only a fraction of the $n^2$ available training pairs per epoch, with fractions ranging from $0.01\%$ to $50\%$. For each fraction, training pairs are sampled randomly at each epoch, and results are averaged over five random seeds.

Despite the quadratic number of possible training pairs, *Rank-Learner* achieves strong ranking performance using only a small subset of the available pairs. In particular, using approximately $1\%$ of the $n^2$ possible pairs already recovers most of the attainable AUTOC, with diminishing returns beyond this point. These results indicate that the computational cost of pairwise training can be reduced substantially without sacrificing performance. Together with the nuisance estimation results in Table 7, this suggests that the overall performance is driven primarily by the quality of the nuisance estimates rather than by extensive pairwise training.

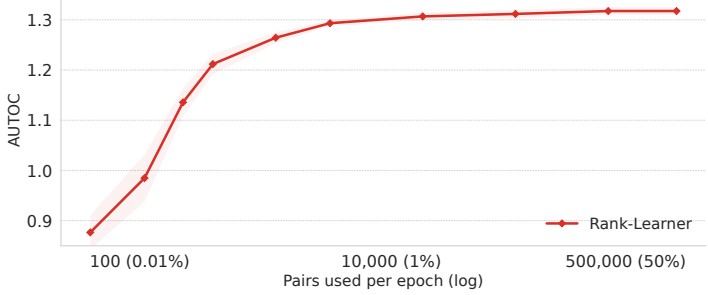

*Figure 5.* **Synthetic benchmark (pair subsampling).** Test AUTOC (mean $\pm$ s.e. over five seeds) of *Rank-Learner* as a function of the number of sampled training pairs per epoch ($n = 1,000$ with $n^2 = 10^6$ possible training pairs). *Higher is better.* The horizontal axis shows the fraction of pairs used per epoch (log).

### F.2.2. ALTERNATIVE PAIR SAMPLING STRATEGY

In addition to uniform pair subsampling, we also evaluate a weighted sampling strategy that prioritizes pairs with larger correction weights. Specifically, pairs are sampled with probability proportional to the weight $\omega_\tau(X, X')$ introduced in Theorem 5.1, which is largest for pairs with more ambiguous plug-in rankings. Table 10 compares weighted and uniform subsampling using 10% of the available training pairs across training sizes and $\kappa$ values. Overall, we observe only minor differences between weighted and uniform pair subsampling across these settings. One possible explanation is that the orthogonal correction in *Rank-Learner* is already pair-dependent through the loss itself, and assigns larger corrections to more ambiguous pairs. These results suggest that uniform subsampling provides a simple and effective strategy in practice.

*Table 10.* **Synthetic benchmark (pair subsampling strategies).** Test AUTOC (mean $\pm$ std dev over five seeds) for weighted and uniform pair subsampling using 10% of the available training pairs across training sizes and $\kappa$ values. *Higher is better.*

| Smoothness parameter | Sampling | $n = 500$ | $n = 1{,}000$ | $n = 2{,}000$ |
|---|---|---|---|---|
| $\kappa = 0.5$ | Weighted | $1.31 \pm 0.02$ | $1.32 \pm 0.02$ | $1.37 \pm 0.00$ |
|  | Uniform | $1.30 \pm 0.01$ | $1.35 \pm 0.00$ | $1.37 \pm 0.00$ |
| $\kappa = 1.5$ | Weighted | $1.29 \pm 0.03$ | $1.32 \pm 0.02$ | $1.36 \pm 0.01$ |
|  | Uniform | $1.29 \pm 0.02$ | $1.33 \pm 0.01$ | $1.36 \pm 0.01$ |
| $\kappa = 3.0$ | Weighted | $1.27 \pm 0.03$ | $1.31 \pm 0.03$ | $1.33 \pm 0.01$ |
|  | Uniform | $1.27 \pm 0.03$ | $1.32 \pm 0.01$ | $1.34 \pm 0.01$ |

### F.3. Robustness and sensitivity analyses

In this section, we study the sensitivity of *Rank-Learner* to overlap violations, the choice of smoothness parameter $\kappa$, clipping of the pseudo labels, and nuisance misspecification.

### F.3.1. OVERLAP SENSITIVITY

Figure 6 studies the effect of decreasing overlap on ranking performance across methods on the synthetic benchmark. In this experiment, we fix the training size to $n = 500$ and vary the degree of overlap by modifying the treatment assignment mechanism, while keeping the remaining components of the synthetic data-generating process fixed (cf. Appendix E.1). Specifically, we replace the propensity score by

$$e(X) = \sigma\Big(\alpha\big(0.8\, s(X) + 0.6(X_6 - 0.5 X_7)\big)\Big), \tag{136}$$

where larger values of $\alpha$ correspond to reduced overlap. We quantify overlap using the empirical mean of $\min\{e(X), 1 - e(X)\}$ over the generated sample. By varying $\alpha$, this overlap measure ranges from approximately $0.5$ (nearly random treatment assignment) to values below $0.05$ (virtually no overlap). As shown in Figure 6, ranking performance degrades smoothly for all methods as overlap decreases. Nevertheless, *Rank-Learner* consistently achieves the strongest performance across the considered overlap regimes, indicating that the proposed ranking objective remains robust under limited overlap.

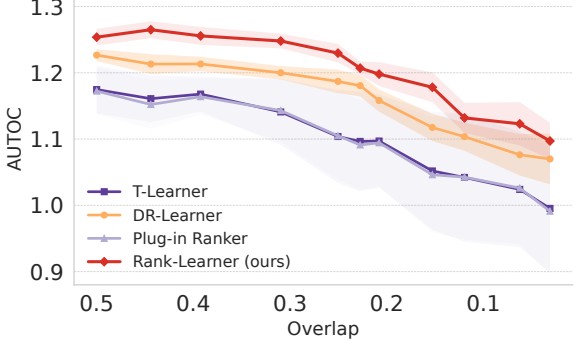

*Figure 6.* **Synthetic benchmark (overlap sensitivity).** Test AUTOC (mean $\pm$ s.e. over five seeds) as a function of overlap (decreasing left to right) for $n = 500$. *Higher is better.* Overlap is varied by changing treatment assignment, while the remaining components of the synthetic data-generating process are kept fixed.

### F.3.2. NUISANCE MISSPECIFICATION

To study robustness to nuisance misspecification, we perform an additional stress test on the synthetic benchmark in which the nuisance functions are deliberately misspecified. Specifically, instead of the neural network nuisance models used throughout the main experiments, we estimate the nuisance components using linear models despite the underlying data-generating process being non-linear. Table 11 reports the resulting AUTOC values for $n = 250$. As expected, all methods degrade under misspecification. However, the degradation is larger for the non-orthogonal methods (T-learner and plug-in ranker) than for the orthogonal methods (*Rank-Learner* and DR-learner).

*Table 11.* **Synthetic benchmark (nuisance misspecification).** Test AUTOC (mean $\pm$ std dev over five seeds) for $n = 250$ under the standard setting and under deliberately misspecified linear nuisance models. *Higher is better.* The oracle column reports the AUTOC obtained by ranking the test set using the true treatment effects. Best mean is shown in **bold**.

| Method | Standard nuisances | Linear nuisances | oracle |
|---|---|---|---|
| T-learner | $0.96 \pm 0.14$ | $0.76 \pm 0.07$ | |
| DR-learner | $1.16 \pm 0.12$ | $1.09 \pm 0.11$ | 1.40 |
| Plug-in ranker | $0.95 \pm 0.14$ | $0.66 \pm 0.06$ | |
| Rank-learner (*ours*) | $\mathbf{1.28 \pm 0.03}$ | $\mathbf{1.19 \pm 0.04}$ | |

### F.3.3. ROBUSTNESS TO LABEL CLIPPING

In our experiments, we clip the orthogonal pseudo labels to the interval $(0, 1)$ for numerical stability, since the orthogonal correction can produce values outside the valid range of the binary cross-entropy loss. Since this modifies the ideal orthogonal objective, we empirically assess whether the resulting loss still exhibits reduced sensitivity to nuisance perturbations. On the synthetic benchmark, where the true nuisances are known, we perturb the nuisance functions around their true values and measure the resulting sensitivity of the second-stage gradient norm. Specifically, for each objective $\mathcal{L} \in \{\mathcal{L}^{\text{soft}}, \mathcal{L}^{\text{orth}}\}$, we compute

$$D_{\mathcal{L}}(\varepsilon) = \frac{\left\| \nabla_\theta \mathcal{L}(g, \eta_\varepsilon) \right\|_2 - \left\| \nabla_\theta \mathcal{L}(g, \eta^0) \right\|_2}{\varepsilon}, \tag{137}$$

where $\eta_\varepsilon$ denotes nuisance perturbations of magnitude $\varepsilon$ around the true nuisances $\eta^0$. Figure 7 shows the resulting gradient sensitivity across perturbation magnitudes $\varepsilon$ and smoothness parameters $\kappa$. The clipped orthogonal objective consistently exhibits substantially lower sensitivity than the corresponding plug-in objective.

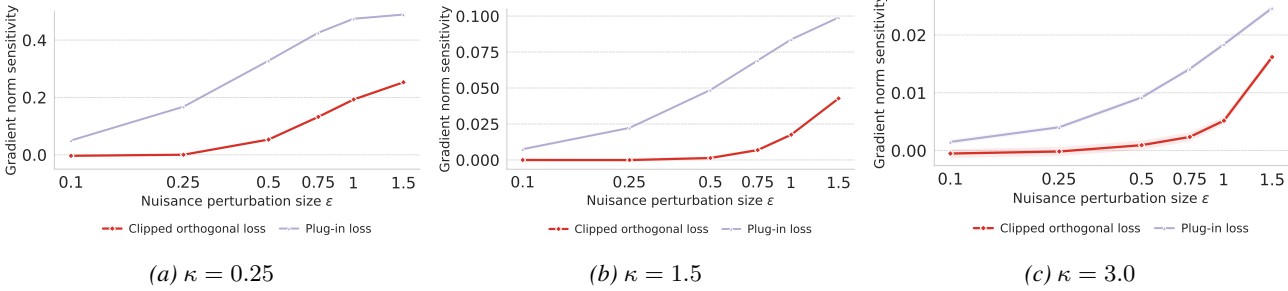

*Figure 7.* **Empirical orthogonality diagnostic.** Sensitivity of the parameter gradient norm of the training objectives to nuisance perturbations around the true nuisances. Across representative values of $\kappa$, the clipped orthogonal objective exhibits substantially lower local sensitivity than the plug-in objective. *Vertical axis not aligned for visual clarity.*

### F.3.4. SENSITIVITY TO THE CHOICE OF $\kappa$

Figure 8 studies the sensitivity of ranking performance to the smoothness parameter $\kappa$ on the synthetic benchmark. We evaluate *Rank-Learner* and the plug-in ranker across a broad range of $\kappa$ values and training sizes. Overall, *Rank-Learner* remains relatively insensitive to the choice of $\kappa$ across the considered values. In particular, for smaller training sizes, *Rank-Learner* consistently outperforms the non-orthogonal plug-in ranker for all considered values of $\kappa$. As the training size increases, performance differences between the methods decrease, which is consistent with improved nuisance estimation and convergence toward oracle performance. These results indicate that the proposed approach is not highly sensitive to the choice of the smoothness parameter in practice.

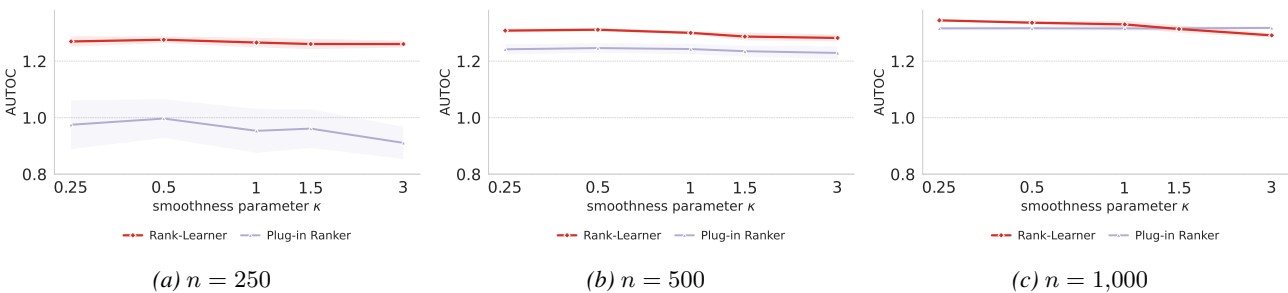

*(a) $n = 250$*      *(b) $n = 500$*      *(c) $n = 1,000$*

*Figure 8.* **Synthetic benchmark ($\kappa$ sensitivity).** Test AUTOC (mean $\pm$ s.e. over five seeds) as a function of the smoothness parameter $\kappa$ across training sizes. *Higher is better*. *Rank-Learner* remains relatively robust to the choice of $\kappa$ over the considered range.

### F.4. Additional comparisons

The primary goal of our main experiments was to isolate the effect of *(i)* direct ranking versus standard CATE estimation and *(ii)* orthogonal versus plug-in objectives for ranking. Therefore, the main experiments focus on baselines that allow us to directly study these empirical questions. In this section, we additionally compare *Rank-Learner* against recent CATE estimation methods and evaluate the proposed method on additional standard causal inference benchmarks.

#### F.4.1. ADDITIONAL BASELINES

Throughout the main experiments, we rely on relatively small neural networks with minimal hyperparameter tuning in order to isolate the empirical effect of direct ranking and orthogonalization. In contrast, the experiments in this subsection are intended to compare against baselines using substantially more sophisticated architectures and training procedures. We have therefore additionally implemented *Rank-Learner* in the same style as the CATENets models (Curth & van der Schaar, 2021), using comparable architectures and optimization procedures for a fair comparison. As a result, the absolute performance levels reported in this subsection differ slightly from those in the main experiments. The corresponding implementation is available in the accompanying code repository.

We compare *Rank-Learner* against several recent approaches for treatment effect ranking and estimation. First, we include the tree-based ranking method of Kamran et al. (2024), which we denote as *tree ranker*. Second, we consider a ranking method based on the DR-learner obtained by constructing soft pairwise ranking targets directly from the doubly robust pseudo outcomes. While doubly robust scores are orthogonal for pointwise CATE estimation, directly inserting them into a ranking loss does not preserve orthogonality. Nevertheless, we include this setting as an empirical baseline, which we denote as *DR ranker*. Third, we compare against recent neural CATE estimation methods from the CATENets library[4], including FlexTENet (Curth & van der Schaar, 2021) and PairNet (Nagalapatti et al., 2024).

Table 12 reports the resulting AUTOC values on the synthetic benchmark. Overall, *Rank-Learner* achieves the strongest performance across training sizes. At the largest training size, the competitive FlexTENet baseline attains marginally higher performance. These results further support the benefits of directly optimizing an orthogonal ranking objective.

*Table 12.* **Synthetic benchmark (additional baselines).** Test AUTOC (mean $\pm$ std dev over five seeds) across training sizes. *Higher is better*. The oracle column reports the metrics obtained by ranking the test set using the true treatment effects. Best mean is shown in **bold**, second best underlined.

| Metric | Method | $n = 100$ | $n = 250$ | $n = 500$ | $n = 1,000$ | $n = 2,000$ | oracle |
|---|---|---|---|---|---|---|---|
| | Tree ranker | $1.012 \pm 0.163$ | $1.200 \pm 0.037$ | $1.273 \pm 0.028$ | $1.303 \pm 0.010$ | $1.338 \pm 0.006$ | |
| | DR ranker | $0.337 \pm 0.544$ | $1.199 \pm 0.090$ | $1.280 \pm 0.010$ | $1.319 \pm 0.017$ | $1.352 \pm 0.008$ | |
| AUTOC ($\uparrow$) | PairNet | $1.028 \pm 0.091$ | $1.194 \pm 0.023$ | $1.242 \pm 0.026$ | $1.265 \pm 0.018$ | $1.279 \pm 0.007$ | 1.40 |
| | FlexTENet | $\underline{1.189 \pm 0.060}$ | $\underline{1.309 \pm 0.015}$ | $1.322 \pm 0.019$ | $\underline{1.345 \pm 0.007}$ | $\mathbf{1.358 \pm 0.011}$ | |
| | Rank-learner (*ours*) | $\mathbf{1.195 \pm 0.129}$ | $\mathbf{1.310 \pm 0.021}$ | $\mathbf{1.326 \pm 0.011}$ | $\mathbf{1.353 \pm 0.008}$ | $\underline{1.358 \pm 0.009}$ | |

---

[4]Available at `https://github.com/AliciaCurth/CATENets`.

F.4.2. ADDITIONAL BENCHMARKS

We additionally evaluate *Rank-Learner* on the ACIC (Dorie et al., 2019) and IHDP (Hill, 2011) benchmarks following the reviewer suggestions. Both benchmarks were originally designed for treatment effect estimation and consist of multiple data-generating processes, some of which exhibit little or no treatment effect heterogeneity and are therefore less suitable for ranking treatment effects. For ACIC, we therefore only consider replications with substantial treatment effect heterogeneity, where treatment effect ranking is meaningful. For IHDP, we report results on the first 50 replications. Since these datasets further contain replications with varying outcome scales, raw AUTOC values are not directly comparable across replications. We therefore report relative AUTOC obtained by min-max scaling within each replication, together with win rates. Table 13 reports the resulting performance. Overall, *Rank-Learner* achieves the strongest relative AUTOC across both benchmarks and competitive win rates.

*Table 13.* **Relative performance on ACIC and IHDP.** We report relative AUTOC, overall win rate, and *Rank-Learner* win rate. Relative AUTOC is obtained by min-max scaling AUTOC within each replication. Overall win rate denotes the fraction of replications in which a method achieves the highest AUTOC, while *Rank-Learner* win rate denotes the fraction of replications in which *Rank-Learner* outperforms the corresponding method.

| Method | ACIC | | | IHDP | | |
| --- | --- | --- | --- | --- | --- | --- |
| | Relative AUTOC | Overall win rate | *Rank-Learner* win rate | Relative AUTOC | Overall win rate | *Rank-Learner* win rate |
| T-learner | 48.1% | 0.00% | 75.0% | 23.1% | 0.0% | 80.0% |
| DR-learner | 47.7% | 12.5% | 87.5% | 62.4% | **42.0%** | 52.0% |
| Plug-in ranker | 53.7% | 12.5% | 75.0% | 50.7% | 18.0% | 68.0% |
| Rank-Learner (*ours*) | **75.6%** | **75.0%** | —- | **66.7%** | 40.0% | —- |

