# OpenReview forum: "Rank-Learner: Orthogonal Ranking of Treatment Effects"
_ICML.cc/2026/Conference — ICML 2026 regular_

### Official Review · Reviewer_Xtg8 · 2026-03-07

**Soundness:** 2
**Presentation:** 3
**Significance:** 2
**Originality:** 2
**Overall Recommendation:** 3
**Confidence:** 4

**Summary:**

This paper studies the problem of ranking individuals according to their treatment effects, a task that arises in many policy and resource allocation settings where it is important to prioritize individuals who are expected to benefit the most from an intervention. A common approach in the literature is to first estimate individual treatment effects and then rank individuals using these estimates. However, such plug-in approaches are known to suffer from bias due to errors in the estimation of nuisance functions, such as the outcome models and propensity scores.

To address this issue, the paper proposes a method that directly learns the ranking functional while correcting for nuisance estimation errors. The authors first construct bias-corrected treatment effect signals using a doubly robust score in the first stage. These corrected signals are then used to generate pairwise binary labels indicating which of two individuals is expected to have a larger treatment effect. The target ranking function is subsequently learned by minimizing a binary cross-entropy loss defined over these pairwise comparisons.

The authors show that the resulting objective is Neyman-orthogonal, meaning that the gradients of the loss with respect to the ranking model are first-order insensitive to errors in the nuisance estimators. As a result, the method inherits desirable theoretical robustness properties. Finally, the paper presents empirical results demonstrating that the proposed approach outperforms their considered baseline methods on some benchmark datasets.

**Compliance With Llm Reviewing Policy:**

Affirmed.

**Final Justification:**

My concerns about the experimental setup remain unresolved: First, the set of baselines is insufficient. Second, the experimental setup is not standardized with respect to prior work. The latter issue could have been partially resolved by reporting results on IHDP; however, the authors instead use CITREO, where confounding is artificially induced in the training data. Overall, the experiments section requires significant revision, and I am therefore not inclined to recommend acceptance. I will maintain my current score.

---

EDIT: I tried running the supplementary code to better understand the IHDP experiment. However, I am upset that I faced several *trivial* issues that suggests that authors have taken a rather casual approach in releasing the code. For instance, there are some trivial syntax errors (e.g., a variable mismatch in ihdp.py, and missing commas in the PairNet implementation).

While I can fix these issues myself, I am concerned that making such modifications on my own may inadvertently lead me to run something different from the authors’ intended implementation. What I am sure of is that the current code is not what the authors used to get the reported (additional) results.

Given this uncertainty, I would prefer **not** to consider any additional experimental results reported on IHDP or ACIC during the rebuttal, and I will retain my score. My concerns remain unresolved.

**Key Questions For Authors:**

I have my main concerns with the experimental section of the paper.

**Synthetic Outcome Functions**

I understand that causal inference research often need to rely on semi-synthetic datasets, but it is generally advisable to use the standard outcome generation procedures associated with benchmark datasets. In this work, the authors seem to introduce their own synthetic outcome functions. This makes it difficult to contextualize the results relative to existing work. It would be helpful to evaluate the proposed method on widely used CATE benchmarks such as IHDP, Twins, and ACIC using the standard outcome generation protocols. For ease of benchmarking against prior work, I'd like to point the authors to the CATENets library (https://github.com/AliciaCurth/CATENets) , as it provides a convenient framework for evaluating many state-of-the-art CATE methods.

**Baselines **

The set of baseline methods considered in the paper is simply not adequate. While the authors include representative approaches such as T-learner and DR-learner, the literature contains many modern implementations and improvements of these methods. Comparing against a broader set of state-of-the-art CATE estimators would provide a clearer picture of the empirical performance of the proposed approach.

In particular, several recent works have proposed strategies to mitigate the limitations of plug-in estimators by focusing directly on improving treatment effect estimation or downstream objectives. Examples include approaches such as FlexTENet https://arxiv.org/html/2305.15984v3 and PairNet (https://arxiv.org/abs/2406.03864), among others. These works emphasize that the accuracy of nuisance function estimation is not necessarily the primary goal, but rather the accuracy of the resulting treatment effect estimates. It would be interesting to compare the proposed method against such approaches and evaluate how it performs relative to these recent developments in CATE estimation.

** Line 306**

The discussion on computational efficiency is misleading. In practice, the proposed approach appears to require substantially more pairwise comparisons in order to rank individuals. If ranking is performed through explicit pairwise evaluations, the number of comparisons could scale on the order of O(n^2), whereas plug-in approaches typically require only O(n) evaluations of the treatment effect estimator (two evaluations per individual). This is conceptually similar to the distinction between early-interaction and late-interaction approaches in ranking systems, where early-interaction methods often incur higher computational cost. Given this, it would be helpful if the authors could clarify the computational complexity of their method during inference. In particular, how does the proposed approach remain computationally efficient when applied to large candidate populations?

**Strengths And Weaknesses:**

Strengths
- The paper is clearly written and well organized, making the technical development easy to follow.
- The motivation for the problem is compelling, particularly in policy settings where identifying individuals with the largest treatment effects is important.
- The theoretical development is presented clearly and the derivations appear intuitive and easy to follow.

Weaknesses / Questions
- Coverage of related work and baselines appears limited. In particular, the paper would benefit from a broader discussion of methods from heterogeneous treatment effect estimation and related ranking approaches.

- While the problem formulation is interesting, the construction of the orthogonal loss appears to follow a standard influence-function–based orthogonalization strategy similar to what is commonly used in the Double Machine Learning literature (e.g., influence-function corrections for nuisance robustness). From the reviewer’s perspective, the main novelty seems to be the application of this orthogonalization framework to pairwise ranking objectives for treatment effect ordering. The authors may wish to clarify whether the theory yields additional insights beyond what is already established in the DML framework.

- The paper relies on Doubly Robust style estimators (e.g., DragonNet) to construct treatment effect labels. However, such approaches are often primarily designed for ATE or outcome modeling settings and may not always perform well for CATE estimation. Even in my own experiments in the past, such methods did not work well for CATE. While the theory is appealing, in practice, I often observed that the propensity models are not well calibrated and hence the DR merits often disappear.

- It would also be interesting to understand whether the plug-in bias is most problematic only when treatment effects are close, i.e., when |τ(x) - τ(x')| is small. For pairs where the difference in treatment effects is large, the ranking should presumably be robust even with imperfect nuisance estimates. Does the theoretical analysis capture or discuss this phenomenon?

- The experimental section is relatively limited, and several aspects would benefit from further clarification or additional experiments.

- The paper would benefit from a discussion of the uplift modeling literature, whose primary goal is also to estimate incremental treatment effects and rank individuals by treatment benefit. Comparing against or situating the proposed method within this literature could strengthen the empirical and conceptual positioning of the work.

---

> ### Author Rebuttal · Authors · 2026-03-30
>
> Thank you for the careful review, we address your main concerns below.
>
> **Empirical evaluation and baselines:** Thank you for your suggestion, based on which **we expanded our evaluation and baselines**. The original experimental design was chosen to answer the paper’s two central questions: (i) direct ranking vs. CATE-then-rank, and (ii) orthogonal vs. plug-in learning. Nonetheless, positioning the method more strongly relative to modern CATE and uplift methods is valuable. As suggested, **we added FlexTENet** as a stronger modern CATE-then-rank baseline, implemented via CATENets, and also included a **DR-based plug-in ranker** as an additional baseline. We find that Rank-Learner performs best among these baselines.
>
> In response to the comment regarding the synthetic evaluation, we **added an experiment on the CRITEO uplift benchmark**. This dataset is based on real-world RCT data from online advertising, and following the benchmark setup, we induce confounding only in the training data, while evaluating on the randomized test data. This complements our original (semi) synthetic experiments with a real-data benchmark that *does not rely on generated treatments or outcomes*. We also checked standard benchmarks (IHDP/ACIC/Twins) and found that they include settings with constant or coarse-grained effect heterogeneity, making them not well suited for ranking (e.g., in Twins the CATE takes only three values). On CRITEO, Rank-Learner performs best among the considered methods in terms of AUUC. The corresponding tables are linked below.
>
> Additional baselines:
> https://anonymous.4open.science/r/review-figures-1D6C/tab2.pdf
>
> CRITEO benchmark:
> https://anonymous.4open.science/r/review-figures-1D6C/tab1.pdf
>
> > **Action:** We will include the experiments with additional baselines and real-world data in the revised paper.
>
> **Computational efficiency:** We would like to clarify that the proposed method does **not** require explicit pairwise comparisons at inference time. The $O(n^2)$ behavior applies *only* during second-stage training, where the pairwise loss is used. The output of training is a scoring function $g(x)$, evaluated once per individual at inference. Individuals are then ranked by sorting these scores, so inference requires only $O(n)$ model evaluations. During training, we mitigate the quadratic cost via uniform subsampling of pairs, and show empirically that performance saturates quickly even when using only a small fraction of all pairs.
>
> > **Action:** We will revise the wording around line 306 to make this clear.
>
> **Positioning and clarifying the scope of contribution:** Thank you for this suggestion. In the revised paper, we will add an extended discussion on uplift modeling and include key works from this literature (e.g, Verbeken et al. (2025), Moraes et al. (2023), Zhang et al. (2021)). The new CRITEO experiment also strengthens the connection of our paper to uplift modelling. The main contribution of our paper is to extend orthogonal learning to the problem of ranking treatment effects. This is non-trivial because the natural ranking objective is *non*-differentiable, so it is **not** directly suited for orthogonalization. To address this, we work with a smooth surrogate, derive a Neyman-orthogonal version of this pairwise loss (Theorem 5.1), and show that its population minimizers preserve the effect ordering (Theorem 5.2). This makes our Rank-Learner the **first** orthogonal two-stage learner for ranking treatment effects.
>
> > **Action:** We will expand the related work and clarify the positioning and contributions in the revised paper.
>
> **Practical robustness of DR-style corrections:** In practice, DR-style methods can suffer from increased variance in finite samples when propensity scores are poorly calibrated or overlap is limited. This is a general challenge of such methods, and requires careful modeling using best-practice principles (e.g., propensity trimming, calibration, etc.). Orthogonality reduces first-order sensitivity to nuisance estimation errors, but still requires reasonably accurate nuisance estimates. Nevertheless, Rank-Learner performs well across our experiments.
>
> **Plug-in bias with small effect differences:** We appreciate this interesting question. Our discussion in the paper is consistent with the intuition that plug-in bias is most problematic when treatment effects are close, since then small nuisance estimation errors can flip the estimated ordering. For pairs with larger effect differences, the ranking is inherently more stable. In our method, the orthogonal correction is pair-dependent and largest for pairs that are hard-to-order under the plug-in estimates (while it becomes small for well-separated pairs). The parameter $\kappa$ further controls how strongly the correction is concentrated on these hard-to-order pairs. We discuss this intuition starting at line 228 (right column).
>
> We thank the reviewer for the detailed comments, which motivated the additions above.

---

> > ### Author Rebuttal · Reviewer_Xtg8 · 2026-04-01
> >
> > 1. Thank you for including the two additional baselines, and for running the CITREO experiments. I agree that in Twins, the outcome is binary, so CATE can only take three values. However, IHDP involves real-valued outcomes. Moreover, unlike CITREO, where confounding had to be artificially induced by the authors; IHDP has built-in confounding, making it a standardized benchmark across prior work. Could the authors please include experiments on IHDP as well? In addition, I believe that PairNet is built on top of the CATENets library (https://github.com/nlokeshiisc/pairnet_release), could you run the CATE-then-rank approach using PairNet and report the results? It would also be helpful if you could point me to the corresponding code.
> >
> > 2. Thank you for the clarification.
> >
> > 3. It is interesting to see that the method extends to uplift modeling settings as well.
> >
> > 4. Thank you for the response.
> >
> > 5. Would it then be a better alternative to sample training pairs based on their weights instead of the current uniform subsampling strategy?

---

> > > ### Author Response · Authors · 2026-04-05
> > >
> > > Thank you for acknowledging the rebuttal and for the follow-up. We have now included **additional experiments on ACIC and IHDP, added PairNet as an additional baseline, added an empirical evaluation of weighted pair subsampling, and extended the anonymous code repository.**
> > >
> > > We included **additional experiments on ACIC and IHDP** and report the results in the table linked here: https://anonymous.4open.science/r/review-figures-1D6C/tab4.pdf. Because ACIC and IHDP consist of heterogeneous replications with different outcome scales, the AUTOC metric is not directly comparable across replications. We therefore report relative performance metrics instead, namely, (1) relative AUTOC (computed by min-max scaling AUTOC within each replication), (2) overall win rate, and (3) the Rank-Learner win rate. For ACIC, we considered replications with high treatment effect heterogeneity, where causal ranking is meaningful. For IHDP, we report results on the first 50 replications. Note that the IHDP dataset is subject to known limitations for causal ML benchmarking and that the results should be interpreted with care (see Curth et al., NeurIPS 2021).
> > >
> > > Under this evaluation, *Rank-Learner is strongest on both ACIC and IHDP in terms of relative AUTOC*. In terms of overall win rate, our Rank-Learner performs best on ACIC, and is tied with DR-Learner on IHDP.
> > >
> > > > **Action:** We will include these additional results in the revised paper.
> > >
> > > We have **added PairNet as an additional baseline** in the updated results: https://anonymous.4open.science/r/review-figures-1D6C/tab2.pdf. Still, *Rank-Learner remains best overall* (while FlexTENet is marginally better on AUTOC at n=500). Note that, since PairNet is not part of the core CATENets library, and the available public implementation is JAX-only, we have implemented a compatible PyTorch version and include it in the anonymous code repository.
> > >
> > > > **Action:** We will include these results in the revised paper.
> > >
> > > Regarding **pair subsampling:** Thank you for this insightful suggestion. We **empirically evaluated a weighted pair sampling strategy**, where pairs that are hard-to-rank based on their plug-in estimates are more likely to be sampled, and report the results here: https://anonymous.4open.science/r/review-figures-1D6C/tab5.pdf. Overall, we observe only a minor difference relative to uniform sampling across training sizes and $\kappa$ values. One possible reason is that the orthogonal correction in Rank-Learner is already pair-dependent, and naturally larger for more ambiguous pairs through the loss itself. Given these results, *uniform subsampling appears to be a simple yet effective strategy, while different pair-subsampling strategies remain an interesting direction for future work.*
> > >
> > > Regarding **code:** We have extended the anonymous repository with code for the additional experiments discussed in the rebuttal. These are provided in the “supplementary” folder alongside the existing code for data generation and for running Rank-Learner and the baseline methods. The repository can be found here: https://anonymous.4open.science/r/rank-learner-v2-55DB
> > >
> > > We thank the reviewer again for the feedback and hope that our response addresses the remaining questions.

---

### Official Review · Reviewer_mnfk · 2026-03-12

**Soundness:** 3
**Presentation:** 4
**Significance:** 3
**Originality:** 2
**Overall Recommendation:** 5
**Confidence:** 3

**Summary:**

The paper introduces Rank-Learners, a two-stage meta-learning approach designed for causal inference when the goal is to determine only the ranking of population's treatment effects rather than their precise Conditional Average Treatment Effect (CATE) magnitudes.

The authors motivate this by arguing that learning the treatment effect's ordering is an easier task to learn. The method operates in two stages:

- Stage One: Estimation of the nuisance parameters, specifically the potential outcomes under treatment $\mu_1(x)$ and control $\mu_0(x)$, and the propensity score $e(x)$ for Inverse Probability Weighting (IPW).

- State Two: Training the final effect estimator $g(x)$ using a novel learning-to-rank loss with smoothed labels.

A key contribution is the proof that their loss function is Neyman--Orthogonal. This property is critical in two-stage meta-learners, as it ensures that the final rank predictions are insensitive (robust) to estimation errors made in the initial nuisance parameter models.

The Rank-Learner is shown to outperform other causal inference methods, such as the T-learner, DR-learner, and plug-in learner, based on the ranking metric of interest, AUTOC.

**Compliance With Llm Reviewing Policy:**

Affirmed.

**Final Justification:**

The rebuttal satisfactorily addresses the main concerns and reinforces confidence in the paper’s technical contributions, particularly the novelty of the causal ranking formulation and the rigorous integration of Neyman-orthogonal learning with pairwise ranking. The proposed framework remains both principled and practically meaningful, with strong empirical results demonstrating clear advantages on ranking focused metrics such as AUTOC.

While some experimental gaps remain especially the absence of a ranking specific DR baseline and additional diagnostic analyses. These do not outweigh the paper’s originality and methodological strength. For these reasons, I am recommending my rating as accept.

**Key Questions For Authors:**

Did the authors test a standard Doubly Robust (DR) learner as a plug-in ranker for comparison? It would be interesting to see if a normal DR model can be adapted with the non-orthogonal $L_{soft}$ loss and compared to the actual orthogonal ranking loss. I wonder if checking this would be a good way to validate empirically the robustness of the proposed method.

**Limitations:**

The method's exclusive focus on ranking limits its utility in high-stakes decision-making. Relying solely on rank overlooks the critical fact that the Conditional Average Treatment Effect (CATE) for certain covariates may be negative; thus, including populations based purely on rank could have negative effects. The paper does not mention a clear, principled cutoff rank beyond which expanding the population for treatment is no longer beneficial.

**Strengths And Weaknesses:**

Strengths

- Novelty of Causal Ranking: Introduces a novel approach to causal inference by focusing on reasoning and decision-making based on the rank of the Conditional Average Treatment Effect (CATE) rather than estimating the absolute CATE value.

- Orthogonal Pairwise Learning: Successfully integrates the principle of Neyman-orthogonal causal learning with a pairwise learning framework, providing rigorous proofs for the orthogonality property.

- Principled Training Mechanism: The method employs a soft ranking surrogate loss and includes a well-reasoned discussion of the bias-variance trade-off associated with tuning the smoothing parameter.

- Strong Empirical Performance: Demonstrates strong performance, particularly in the ranking metric (AUTOC), when compared against other established meta-learner baselines.

Weaknesses

- Missing Baseline Comparison: Lacks a comparison against a strong ranking-specific baseline, such as a Doubly Robust (DR) learner configured as a plug-in ranker.

- Empirical Justification of Task Ease: The current experimental comparison of the learned ranking function g(x) against the actual treatment effect T(x) does not clearly justify the claim that the model is learning a fundamentally "easier decision function."

- Missing Diagnostic Plots for : The paper would benefit from diagnostic plots that show the effect of the smoothing parameter  on the learned ranking function g(x) by comparing it against the true treatment effect T(x).

---

> ### Author Rebuttal · Authors · 2026-03-30
>
> Thank you for the review and for the positive overall assessment. We address your main questions below.
>
> **Extending the empirical evaluation:** The original experimental design was intended to isolate the two main empirical questions of the paper: *(i)* ranking vs. regression and *(ii)* orthogonal vs. plug-in learning. We appreciate your suggestion to include a comparison with stronger baselines. Following your suggestion, **we therefore added** a (1) **plug-in ranker** based on DR-Learner estimates, and also included (2) an additional modern **CATE-then-sort baseline based on FlexTENet**. Both methods are strong baselines, but neither directly orthogonalizes the ranking objective itself. We find that our *Rank-Learner remains strongest overall in AUTOC*, which provides additional empirical support for the proposed method. To further strengthen the empirical evaluation, we also performed an **additional experiment** on the CRITEO uplift dataset, which is based on real-world RCT data, where Rank-Learner performs best among the considered methods. The corresponding results can be found in the links below.
>
> CRITEO benchmark:
> https://anonymous.4open.science/r/review-figures-1D6C/tab1.pdf
>
> Additional baselines:
> https://anonymous.4open.science/r/review-figures-1D6C/tab2.pdf
>
> > **Action:** We will include the experiments with additional baselines and real-world data in the revised paper.
>
> **Diagnostic plot for kappa:** Thank you for this suggestion, we added additional diagnostics for the smoothing parameter $\kappa$. Since the goal of Rank-Learner is to recover the ordering of treatment effects rather than pointwise CATE magnitudes, we believe the most relevant diagnostic in our setting is how $\kappa$ affects downstream ranking quality. We therefore performed an additional sensitivity analysis over $\kappa$ and found that downstream ranking performance remains robust across a broad range of values.
>
> Sensitivity analysis ($\kappa$) : https://anonymous.4open.science/r/review-figures-1D6C/fig1.pdf
>
> > **Action:** We will include this figure in the revised version.
>
> **Empirical justification of task ease:** We would like to clarify that our claim that ranking is easier than CATE estimation is mainly about the learning target itself. A pointwise regression objective is minimized *only* by the true CATE function, while a ranking objective only requires recovering the correct treatment effect ordering. Thus, *many scoring functions can be equally optimal for ranking as long as they preserve the ordering, while regression imposes a much stronger requirement*. Empirically, this is reflected in the fact that directly optimizing for ranking leads to better performance than CATE-then-rank baselines on AUTOC.
>
> > **Action:** We will clarify this in the discussion starting at line 185 (right column).
>
> **Negative treatment effects:** We appreciate this interesting point. It is true that ranking alone is not sufficient to determine a stopping point once treatment effects become negative. Our method is designed for ranking, where the downstream objective depends *only* on the ordering, and not the treatment effect magnitudes. Selecting a treatment cutoff is a related but different problem, since it also requires magnitude information near the decision boundary. This related setting has received some prior attention (see e.g.  Frauen et al., NeurIPS 2025).
>
> > **Action:** We will make this distinction clear in the revised paper.
>
> We thank the reviewer for the positive assessment of the paper and the suggestions, which helped us improve the paper.

---

> > ### Author Rebuttal · Reviewer_mnfk · 2026-04-03
> >
> > Thank you for the comprehensive rebuttal and for addressing my questions. I will maintain my rating of 'accept'.

---

> > > ### Author Response · Authors · 2026-04-05
> > >
> > > Thank you for the positive feedback. We are glad that the rebuttal addressed the remaining questions.

---

### Official Review · Reviewer_y2oF · 2026-03-13

**Soundness:** 3
**Presentation:** 3
**Significance:** 2
**Originality:** 3
**Overall Recommendation:** 5
**Confidence:** 4

**Summary:**

Overall Review
The paper introduces an orthogonal two-stage learner for ranking individuals by treatment effects, rather than estimating effect magnitudes and then ranking individuals.

The core insight is that ranking is an easier target than pointwise estimation and should be optimized directly - this is interesting and relevant. The technical novelty is moderate given the existing orthogonal learning toolkit. The experimental evaluation relies entirely on synthetic/semi-synthetic benchmarks. Weak accept.

Summary
Many downstream decisions require only a ranking of who will respond, not their magnitudes. Standard practice estimates CATEs and then sorts. The authors propose Rank-Learner, which directly optimizes a pairwise ranking loss constructed from observational data. The key technical contribution is deriving a Neyman-orthogonal version of this ranking loss. The method follows a two-stage approach: estimate nuisances (propensity scores) via cross-fitting, then use that to generate the loss function. Experiments on synthetic and semi-synthetic benchmarks show consistent improvements over CATE-then-rank baselines and non-orthogonal rankers.

**Compliance With Llm Reviewing Policy:**

Affirmed.

**Final Justification:**

Authors did 2 new experiments to address two of the comments.

**Key Questions For Authors:**

See my weaknesses above. My main concern is that the contribution is somewhat thin for an ICML paper. Two questions which I believe may strengthen the paper are:

1) Is ranking more robust to misspecification/missing covariates than CATE-then-rank? It seems like it may be but I am un sure.
2) If we extend this to continuous valued or multi-treatment experiments, does the advantage of rank vs. CATE-then-rank grow or shrink?

**Limitations:**

yes

**Strengths And Weaknesses:**

Strengths
The central observation that ranking is easier than estimation is well motivated and interesting

The experimental design does all ablations (i.e. direct ranking vs. CATE-then-sort, and orthogonal vs. plug-in objectives)

The method is model-agnostic and the orthogonal correction enters only through the score estimates meaning the second-stage training problem is standard pairwise ranking.

Weaknesses

The technical contribution appears to be moderate. I am not a very deep expert in this particular field, but the derivation appears to be standard.

All experiments use synthetic or semi-synthetic outcomes. This is standard in causal inference but limits the persuasiveness of the results.
A real observational dataset where ground-truth rankings can be partially validated (e.g., through a companion RCT) would substantially strengthen the paper.

The practical motivation emphasizes observational data settings (healthcare, marketing, policy), but the experimental setup uses clean, well-specified confounding mechanisms with known functional forms. It would be valuable to stress-test under model misspecification – e.g., what happens when the nuisance models are not just noisy but structurally wrong and/or important covariates are not observed.

Other Notes
Only the binary treatment case is considered. Many of the motivating applications involve multi-valued or continuous treatments (dosing decisions, variable ad spend), and extending the pairwise framework to these settings is non-obvious and would certainly increase the technical contribution of the paper.

---

> ### Author Rebuttal · Authors · 2026-03-30
>
> Thank you for the review, and for highlighting both the practical motivation for direct ranking and the value of the experimental ablations. We address your main concerns below.
>
> **Evaluation on real-world data:** Thank you for this helpful suggestion to include an additional real-world dataset. In response, we performed **an additional experiment on the CRITEO uplift dataset** from a marketing context. The underlying data comes from a true RCT, and following the original benchmark setup, we induce confounding only in the training data (via selective subsampling of training instances), while evaluating ranking performance on randomized test data. This provides an additional realistic benchmark, where we evaluate with the AUUC metric (area under the uplift curve). In this experiment, *Rank-Learner performs best among the considered methods*, providing additional empirical support beyond the synthetic and semi-synthetic experiments. The corresponding table can be found below.
>
> CRITEO benchmark:
> https://anonymous.4open.science/r/review-figures-1D6C/tab1.pdf
>
> > Action: We will add the additional real-world data experiment in the revised paper.
>
> **Robustness to misspecification and unobserved confounding:** Regarding model misspecification, orthogonality is designed to reduce first-order sensitivity to nuisance function errors, regardless of whether these errors arise from finite-sample estimation or model misspecification. In this sense, Rank-Learner is expected to be more robust than CATE-then-rank whenever the nuisance functions are estimated imperfectly, regardless of the source of that error. To assess this empirically, we performed a small additional stress test with deliberately misspecified linear nuisance models, *which supports this expectation* (the corresponding table is linked below). However, orthogonality is still a local robustness property, so it relies on the nuisance estimates being reasonably accurate, severe misspecification cannot be resolved by orthogonality alone.
>
> Missing important covariates corresponds to unobserved confounding, which is a different problem setting. Both Rank-Learner and CATE-then-rank rely on standard identifiability assumptions, so neither can generally recover the correct treatment effect ordering under hidden confounding without additional assumptions. This is also an active research area in its own right in causal inference and policy learning. Extending methods such as Rank-Learner to the setting of hidden confounding is an interesting direction for future work.
>
> Nuisance misspecification:
> https://anonymous.4open.science/r/review-figures-1D6C/tab3.pdf
>
> > Action: We will clarify the scope of orthogonality in the discussion starting at line 161 and add hidden confounding as an important direction for future work.
>
>
> **Technical novelty:** The paper builds on ideas from semiparametric and orthogonal learning, and extends them to the problem of ranking treatment effects. Our extension is not straightforward and highly non-trivial, because, unlike many other causal ML papers, *the target is no longer pointwise estimation of CATE magnitudes, but the direct learning of a score function for the treatment effect ordering.* The natural ranking objective is *non*-differentiable and therefore **not** directly suited for orthogonalization, which requires working with a smooth surrogate, deriving a Neyman-orthogonal objective (Theorem 5.1), and showing that the population minimizers preserve the correct ranking (Theorem 5.2). This makes Rank-Learner the **first** orthogonal two-stage learner designed for ranking treatment effects.
>
> **Extensions to multi-valued and continuous treatments:** We view the considered binary treat-versus-no-treat setting as a good starting point and one that is relevant for many practical applications. Extending the framework to multi-level or continuous treatments would be valuable, but is not straightforward. For instance, in the multi-level treatment case, we must first decide what should be ranked (relative to what treatment level), which changes the nuisance structure, the loss function and the orthogonal correction. We thank the reviewer for this interesting suggestion.
>
> > Action: We will add this extension explicitly as an important direction for future work.
>
> We thank the reviewer again for the helpful feedback, which helped us strengthen the paper.

---

> > ### Author Rebuttal · Reviewer_y2oF · 2026-03-31
> >
> > I thank the authors for this well structured response to my review and the extra experiments.  I am willing to change my rating up to Accept from Weak Accept.

---

> > > ### Author Response · Authors · 2026-04-05
> > >
> > > Thank you for your constructive review and for acknowledging our rebuttal and additional experiments. We noticed the score update (Accept) is not yet reflected in the system and wanted to check if there is anything to do on our side (or whether this is an issue with OpenReview).

---

### Official Review · Reviewer_j6o1 · 2026-03-14

**Soundness:** 3
**Presentation:** 3
**Significance:** 3
**Originality:** 3
**Overall Recommendation:** 4
**Confidence:** 4

**Summary:**

The authors introduce 'Rank-Learner,' a two-stage Neyman-orthogonal framework for causal ranking. The approach posits that point-estimation of the Conditional Average Treatment Effect (CATE), $\tau(x)$, is often a 'harder than necessary' task when the decision-making goal is limited to prioritization. By optimizing a differentiable, pairwise surrogate loss that satisfies Neyman-orthogonality, the framework ensures that the scoring function $g(x)$ remains robust to first-order nuisance estimation errors. Empirical results, evaluated via the Area Under the Threshold-varying Operational Characteristic (AUTOC), demonstrate consistent performance improvements over competing learners across various observational scenarios.

**Compliance With Llm Reviewing Policy:**

Affirmed.

**Final Justification:**

I thank the authors for providing additional sensitivity experiments. While these results are helpful, other concerns remain unaddressed and are deferred to future research. I acknowledge the complexity of these tasks, but given the remaining gaps, I maintain my original recommendation.

**Key Questions For Authors:**

1. **Regret Analysis:**
Can you derive, or at least hypothesize, how the "hardness" reduction translates into a formal regret bound for the AUTOC metric compared to other learners?
2. **Optimal Thresholding:**
Is this framework strictly limited to ranking, or can it be extended to determine an optimal decision threshold (e.g., a cost-benefit cutoff)?
Specifically, consider a scenario with two sets of CATE values: $S_1 = \{100, 4, 3, 2, 1\}$ and $S_2 = \{100, 99, 3, 2, 1\}$. While the *rank-learner* may correctly order both sets, the optimal decision point changes significantly depending on the underlying magnitude gaps. For instance, in $S_1$, a cutoff might naturally isolate $100$ from the rest, whereas in $S_2$, $100$ and $99$ are grouped together as a high-impact cluster.
Given that your framework optimizes for relative ordering rather than absolute magnitude, how might the orthogonalized objective be adjusted if the decision-maker needs to identify these "gap-based" thresholds rather than simply performing a top-$K$ selection? Do you view this as a fundamental limitation of pure ranking, or is there a way to adapt your orthognalization-based approach to preserve magnitude information at the decision margin?

**Limitations:**

* **Computational Complexity:** The pairwise nature of the objective function inherently scales at $O(n^2)$, which poses a significant bottleneck for large-scale production datasets. While empirical results in Figure 3 demonstrate that subsampling training pairs—using only a small fraction of the total possible pairs per epoch—maintains strong AUTOC performance, this practice lacks theoretical justification.
* **Heuristic Subsampling:** Given the success of subsampling, there is a clear need for a more principled randomization scheme in the second stage. Current practice relies on empirical observation rather than a formal analysis of how pair-selection strategies affect the convergence of the Neyman-orthogonal estimator.

**Strengths And Weaknesses:**

### **Strengths**

* **Principled Robustness:** The use of Neyman-orthogonality to construct the ranking objective is an excellent application of semiparametric theory to a domain (ranking) that typically ignores nuisance parameter propagation.
* **Task Alignment:** The argument that pointwise regression for $\tau(x)$ is "harder than necessary" for ranking is well-taken and aligns with the practical requirements of resource-constrained decision-making.
* **Methodological Clarity:** The two-stage procedure—estimating nuisance parameters and subsequently optimizing an orthogonalized pairwise loss—is intuitive and flexible regarding model architecture.

### **Weaknesses**

* **Lack of Theoretical Regret Bounds for AUTOC:** While the objective function is proven to be orthogonal, the paper does not derive formal convergence bounds for the AUTOC metric itself. Consequently, the link between objective-level orthogonality and ranking-level optimality remains largely empirical.
* **Heuristic Components:** The reliance on the hyperparameter $\kappa$ (for the soft-ranking loss) and pseudo-label clipping is somewhat ad-hoc. The paper lacks a rigorous discussion regarding how these choices affect the final ranking stability or the theoretical validity of the orthogonalization under finite-sample constraints.

---

> ### Author Rebuttal · Authors · 2026-03-30
>
> Thank you for the constructive review, and for recognizing the orthogonality application, the task alignment, and the overall framework. We address your main questions below.
>
> **Link between orthogonality and ranking optimality:** We currently do not provide a formal regret bound for AUTOC, and agree this would be a valuable addition. At the same time, the current theory does establish a population-level link between the proposed objective and ranking quality. Theorem 5.1 shows that the objective is Neyman-orthogonal, while Theorem 5.2 shows that its population minimizer preserves the true treatment effect ordering. Since AUTOC depends only on the induced ordering, *these results connect objective-level orthogonality to ranking-level optimality, at the population level*. A formal AUTOC regret analysis is challenging because AUTOC is a global ranking metric, so we need to relate ranking errors, including those from nuisance estimation, to the final metric. We view this as an interesting direction for future work, and thank the reviewer for this suggestion.
>
> > **Action:** We will clarify this population-level connection in the revised paper and discuss formal AUTOC regret bounds as an important direction for future work.
>
> **Clarification on kappa and clipping:** We would like to clarify that $\kappa$ is the smoothing parameter of the differentiable pairwise ranking objective that enables orthogonalization and controls a bias-variance trade-off in our method. We agree, however, that the original submission did not thoroughly discuss its impact on downstream ranking performance. To address this, we performed **an additional sensitivity analysis** showing that performance remains robust across a broad range of $\kappa$ values (see the link below for the corresponding figure).
>
> Regarding clipping, we appreciate the opportunity to explain its role more explicitly. Clipping is included to ensure stable optimization in finite samples when nuisance estimates are noisy. To understand its impact on the benefits of orthogonalization, we performed **an additional empirical diagnostic** on synthetic data: we measured how sensitive the second-stage gradients are to perturbations of the nuisances around their true values. The results show that the clipped orthogonal objective remains substantially less sensitive to such perturbations than the corresponding plug-in objective (see the link below for the corresponding figure), *which confirms the effectiveness of the proposed approach*.
>
> Sensitivity analysis ($\kappa$) : https://anonymous.4open.science/r/review-figures-1D6C/fig1.pdf
>
> Empirical orthogonality: https://anonymous.4open.science/r/review-figures-1D6C/fig2.pdf
>
> > **Action:** We will add a more detailed analysis on the role of clipping, as outlined above, to our revised paper.
>
> **Optimal thresholding:** We appreciate this insightful question, which highlights an important distinction between ranking and threshold-based decision-making. Our framework is designed for ranking problems, where the downstream objective depends *only* on ordering rather than treatment effect magnitudes. Threshold-based decision-making is a related but different objective, since it also requires preserving magnitude information near the decision boundary (this related problem has been studied previously, see e.g. Frauen et al., NeurIPS 2025).
>
> > **Action:** We will clarify this distinction explicitly in the paper.
>
> **Computational complexity and pair subsampling:** The pairwise objective is quadratic in $n$, so we used uniform random subsampling of pairs as a practical scalability mechanism. A theoretical analysis of how this subsampling affects the second-stage estimator and downstream AUTOC performance would be valuable, as would the design of more principled pair-selection strategies.
>
> > **Action:** We will explicitly add this as important directions for future work.
>
> We thank the reviewer again for the comments and suggestions, which helped us improve both the presentation and the empirical part of the paper.

---

> > ### Author Rebuttal · Reviewer_j6o1 · 2026-04-03
> >
> > I thank the authors for providing additional sensitivity experiments. While these results are helpful, other concerns remain unaddressed and are deferred to future research. I acknowledge the complexity of these tasks, but given the remaining gaps, I maintain my original recommendation.

---

> > > ### Author Response · Authors · 2026-04-05
> > >
> > > Thank you for acknowledging our rebuttal. We appreciate your careful review and the helpful feedback.

---

### Decision · Program_Chairs · 2026-04-30

**Decision:**

Accept (regular)

**Comment:**

Reviewers agreed that the main novelty of the work is that the authors were able to establish a neyman orthogonal loss for ranking causal effects. A few notable concerns that the reviewers raised were:
1. robustness to the smoothing parameter and the pseudo label clipping. The authors conducted sensitivity analysis that addresses these concerns
2. Lack of evaluation on real world/additional data. The authors added analysis on the CRITEO benchmark + ACIC and IHDP
3. robustness to model misspecification: the authors clarified that Neyman Orthogonality is particularly useful in this setting and empirically showed that robustness to nuisance model misspecification holds
4. missing baseline models: the authors included additional baselines and reported their performance